

# Using paraxial approximation to describe the optical setup of a typical EARLINET lidar system

Panagiotis Kokkalis[1]

[1]Institute of Astronomy, Astrophysics, Space Applications and Remote Sensing, National Observatory of Athens, 15236, Greece

*Correspondence to*: Panagiotis Kokkalis (panko@noa.gr)

**Abstract**

The mathematical formulation for the optical setup of a typical EARLINET lidar system is given here. The equations describing a lidar system from the emitted laser beam to its projection on the final receiving unit are revealed, based on paraxial approximation and geometric optics approach. The evaluation of the formulation is performed with ray tracing simulations, revealing an overall good performance with relative differences of the order of 5% mainly attributed to aberrations that are not taken into account in paraxial approximation.

## 1 Introduction

Lidars are efficient tools for retrieving the aerosol optical and microphysical properties in the Planetary Boundary Layer (PBL) and free troposphere (FT). More precisely, the lidar techniques that are widely used for aerosol research, are capable of providing range-resolved information for: (a) the aerosol backscatter coefficient ($\beta_{\alpha er}$), through the backscatter lidar technique (e.g. Fernald et al., 1972; Klett, 1981); (b) the aerosol extinction coefficient ($\alpha_{\alpha er}$), through the Raman lidar technique (Ansmann et al., 1990; Ansmann et al., 1992) and (c) the volume and particle linear depolarization ratios ($\delta_v, \delta_{\alpha er}$), through the depolarization lidar technique (e.g. Sassen, 2005; Freudenthaler et al., 2009). Many studies have demonstrated that the provision of the aforementioned aerosol optical properties for multiple wavelengths facilitates the retrieval of aerosol microphysical properties through inversion techniques (Müller et al., 1999; Veselovskii et al., 2002; Veselovskii et al., 2010). The accuracy of the optical properties used as inputs for the inversions is critical. Uncertainties involved for the retrieval of the aerosol optical properties with lidar techniques are linked both to systematic and statistical sources of error. Statistical errors are mainly attributed to the signal detection itself and the error introduced by operational procedures within the data processing chain, such as signal averaging during varying atmospheric extinction and scattering conditions (Ansmann et al., 1992; Bösenberg, 1997; Iarlori et al., 2015). Systematic errors are mostly linked to the estimation of temperature and pressure profiles along with the wavelength dependence parameter required in Raman



technique (Ansmann et al., 1990; Whiteman, 1999). During the last decades, a lot of work towards estimating and minimizing the errors in aerosol lidar retrievals has been done in the framework of the European Aerosol Research Lidar Network (EARLINET; Pappalardo et al., 2014). For example and in order to optimize the optical performance and control the quality of aerosol measurements, a number of Quality Assurance (QA) tests has been adopted and applied in EARLINET

lidar systems (Freudenthaler, 2008). Moreover, an increased effort has been put by the European lidar community, to develop and apply accurate depolarization calibration techniques (Freudenthaler, 2016) and quantify and correct the influence of systematic error introduced by imperfections of lidar optical elements on the depolarization related retrievals (Mattis et al., 2009; Bravo-Aranda et al., 2016; Belegante et al., 2016). These studies are based on the description state of polarization of light and lidar optical elements, by means of the Müller-Stokes formulation. The current study is

complementary to the aforementioned work, since it is also linked with the detected lidar signal, however without being able to provide any information related to depolarization retrievals, due to the usage of paraxial optics formalism.

The lidar equation in its simplest form includes the overlap function ($O(z)$) and the overall optical efficiency of the system. The overlap function is range-dependent and thus related to the lidar system geometry, since it describes the fraction of the laser beam cross section contained within the receiver field of view, taking values from 0 to 1 (Wandinger, 2005). At the

height range where the overlap function reaches the value of 1, the laser beam divergence (TFOV) and receivers field of view (RFOV) are fully overlapped. This range is denoted as distance of full overlap (DFO) and is usually extended from 500 to 1500 m for EARLINET aerosol lidar systems. The overlap function and the "blind" region of DFO are adding a significant drawback on the retrieval of aerosol optical properties from lidar systems, since it becomes difficult to obtain useful and accurate information regarding the aerosol entrapped below that height range. More precisely, according to Wandinger and

Ansmann (2002), the uncertainties regarding aerosol extinction coefficient are up to 200%, for heights below DFO, making the retrievals not useful up to that height range.

Thus, in order to optimize the performance of a lidar at lower altitudes and effectively retrieve optical properties of the aerosol entrapped below the PBL height it is of great importance that the receiving telescope is able of detecting the emitted laser pulse, already at short ranges from the lidar system. Therefore, a low full overlap height is needed. The wide angle

RFOV is not an optimal solution for minimizing the DFO, since in that case: (a) the signal will be contaminated with more sky background light, and (b) multiple scattering effects have to be taken into account especially for cases where non-spherical particles are suspended in the atmosphere (Wandinger, 1998; Wandinger et al., 2010).

Case studies, but also long-term lidar observations performed during the last decade at various EARLINET stations, revealed that the DFO have to be much lower than 600 m in order to detect the boundary layer at European latitudes, especially during

winter time (e.g. Matthias and Bösenberg, 2002; Matthias et al., 2004; Amiridis et al., 2007; Baars et al., 2008). Measurements of the aerosols, within the PBL are in particular required during daytime, when the convection is stronger. However, daytime lidar operation suffers from the increased sky radiance contaminating the lidar signal, which needs suppression. In order to suppress the bright day time sky radiance and enhance the Signal to Noise Ratio (SNR) of the lidar signal, small bandwidth ( BW ) IFF are widely used in EARLINET (http://www.meteo.physik.uni-



muenchen.de/~stlidar/earlinet_asos/hoi/EA-NA4-HOI-overview-frames.html). Such IFFs have recently become commercially available with small BW of the order of 0.17 nm (FWHM) at visible spectrum (Alluxa). A significant drawback of these filters is that a decrease bandwidth can be caused when the acceptance angle ($A_{IFF}^{max.}$) is decreased as well, which in turn limits the possible DFO. Alternative methods for efficiently suppressing the background are based on the

shaping of the receivers field of view diaphragm (FOVD) along with their geometry and their relative position on the optical axis, as has been proposed by Abramochkin et al. (1999) and Freudenthaler (2003).

There are several studies in the literature, related to the determination of the overlap function of lidar systems analytically (e.g. Jenness et al., 1997; Chourdakis et al., 2002; Stelmaszczyk et al., 2005; Comeron et al., 2011), or experimentally (e.g. Sasano et al., 1979; Tomine et al., 1989; Dho et al., 1997; Guerrero-Rascado et al., 2010; Wandinger and Ansmann 2002; ).

For the theoretical approaches, a good understanding of the actual light distribution in the laser beam cross section, and the characteristics of the receiving unit are needed to obtain an overlap profile with sufficient accuracy. Stelmaszczyk et al., (2005), proposed an analytical formula also for decreasing the DFO based only on the laser-telescope geometry and specifically the introduction of a small inclination between the transmitter and receiver central axis. However, all the aforementioned theoretical studies provide information regarding the overlap function based explicitly on the laser-telescope

geometry.

This study focuses on the extension of the paraxial approximation down to the detector, revealing all the possible constraints of a lidar setup, since DFO depends on the overall optical path of the detected backscattered radiation. The distance of full overlap, as presented in this work, depends on the entire geometry considering all the parts of the lidar as one optical system. The analysis in this study highlights the need to take into consideration the acceptance angle of the interference filter when

designing an optimized lidar system and the possible limitations that this imposes. The corresponding geometrical formulation is presented in Section 2, describing the basic characteristics (focal lengths, distances, and diameters) of all the optical components, compromised with the EARLINET QA standards. Results of ray tracing simulations with respect to lidar design and alignment according to geometrical formulation are presented in Section 3. In Section 4 we present the influence of mounting accuracy on the final DFO, and in Section 5 the geometrical characteristics of the field stop, through

paraxial approximation formulas.

The entire set of equations, has been integrated in a Microsoft Excel Worksheet through Visual Basic for Application (VBA) code, and distributed with supplementary documentation to the members of EARLINET network (http://www.meteo.physik.uni-muenchen.de/~stlidar/earlinet_asos/raytracing/Basic_design/basic_lidar_design.html). The aforementioned worksheet cannot substitute advanced and expensive optical design software. However, it may act

complementary, for the preliminary design of a system, or can be used as a quick check up tool of an existing lidar system, or even finally as a learning tool for becoming familiar with an optical lidar setup.



## 2 Lidar optical setup and limitations

Lidar systems are using large telescopes to collect the weak light, backscattered from the atmosphere. This portion of light, has to be further transmitted and projected, without any range dependent losses, to small detectors. For example, with a telescope diameter of 300 mm and a detector diameter of 5 mm, an overall magnification of the optical system of 0.0166 is

necessary. On the other hand, the angular magnification is increased by 60, which means that 1.25 mrad field of view of the telescope, is magnified to about 75 mrad (~ 4.3°).

For single but more particularly for multi-wavelength systems, the wavelength separation unit is not capable to accept such a divergent beam, since: (a) it would be soon too wide for 1" or even 2" optical elements; and (b) the transmission of interference filters is very sensitive to the incidence angle. Therefore the magnification of the receiver optics is split in two

parts, i.e. the telescope with a collimation lens, and another objective with an eyepiece, with a low divergent light path ("parallel beam") in between. The divergence of the "parallel" beam is determined by the laser divergence, the laser-telescope axis distance, the tilt of the laser beam with respect to the telescope axis, determining the field of incidence angles into the telescope, and the magnification of the telescope together with the collimating lens.

The limitations for this divergence are the following: the field of view of the telescope should be as small as possible in order

to reduce the background light collected from the sky; the beam diameter and its divergence, must both be small enough to fit through the 1" optics for all necessary beam splitters; the divergence must be less than the maximum acceptance angles of the interference filters.

### 2.1 Lidar optical layout

The setup of a biaxial lidar system is schematically given in Fig. 1. The laser-telescope geometry is demonstrated in the

upper part (Fig. 1a) while the optical setup of the lidar receiving unit behind the telescope, is presented in the lower part (Fig. 1b). The abbreviations used in this study in order to describe the lidar parameters, are summarized in Table 1.

An ideally circularly shaped laser beam with initial diameter DL and divergence TFOV (half angle) is transmitted in the atmosphere. The laser beam interacts with the atmospheric constituents (aerosols and molecules) and the backscattered light is collected by a telescope, with a focal length FT and clear aperture DT. The distance between the transmitter and receiver

central axis considered to be equal to DTL (Fig. 1a). The RFOV (half angle) of the telescope is determined by a diaphragm FOVD (usually a circular iris), with diameter $D_{FS}$ centered on the optical axis, and mounted after the telescope.

For simplicity, all the optical components of the system (telescope and lenses) are presented like thin lenses in Fig. 1b. Moreover, the paraxial approximation approach implies that aberration effects due to non-ideal performance of the optical parts are not considered (i.e. chromatic aberration, focal blur of the telescope, spherical aberrations of the collimator and

following lenses).

Initially, the rays collected by the telescope are coming both from far (parallel to the optical axis) and near range (with an inclination determined by the RFOV); (green and blue lines in Fig. 1b respectively), focused on its focal plane and thus



spatially filtered by FOVD. The diverging beam must be collimated by a first lens (collimator) with diameter $D_{col}$ and focal length $F_{col}$, mounted at distance $F_{col}$ behind the field stop. The collimation of the near range rays is mandatory due to the limited acceptance angles $A_{IFF}^{max.}$ of the IFFs (see appendix). By collimating the far and near range rays, the collimating lens produces an intermediate image (II) of the entrance pupil at a distance $Z_{II}$ behind the lens. At the so called eye-relief plane, the far and near range rays are crossing each other and in that case if an optical detection device is mounted there, it is feasible to obtain the full viewing angle. In the optical setup presented here, this intermediate image appears twice; first behind the collimator and secondly behind the eyepiece. At this point the image projected by the telescope becomes sharper and is independent on the lidar range. An objective lens (focal length $F_{obj}$, diameter $D_{obj}$) is located just behind the IFF and at a distance $Z_1$ behind the collimator. An eyepiece lens (focal length $F_{eye}$, diameter $D_{eye}$) is placed at a distance of $Z_2 = F_{obj} + F_{eye}$, behind the objective lens. The focal length of the objective lens must not be shorter than 3 times its diameter, in order to avoid excess spherical aberrations. This is a critical parameter since for low $F_{obj}$, of a simple plano-convex or bi-convex lens, there is an increased risk that the image of the telescope aperture on the photomultiplier (PMT), to be unstable with lidar range. The PMT with diameter $D_{PMT}$, is located at distance $Z_3$ behind the eyepiece lens and on its surface has to be projected the image detected by the clear aperture of the telescope. The aforementioned components have to fulfill specific conditions, regarding their diameter and focal length, and should be accurately mounted on the optical path of the collected backscattered light, in order to achieve a sufficient imaging of the telescope's aperture onto the detector's effective surface.

In principal, the image of the laser beam (in object space) that is collected by the telescope's clear aperture (values above 200 mm) have to be projected on the effective diameter of the photomultiplier, maximum 8 mm for Hamamatsu PMTs R7400 series (Hamamatsu Photonics, 2006), (in image space). The useful diameter of the PMT is about 5 mm, including mounting and adjustment tolerances.

For ranges above the DFO, the laser beam stays entirely inside the telescope's full field of view. For those ranges, the extreme points of the telescope mirror and consequently each point of the telescope, detect the laser pulse entirely and with the same collecting efficiency. This is true for small inclination angles of the laser central axis towards the telescope axis ($A_{tilt}$).

## 2.2 Description through paraxial approximation

The parameter of RFOV is chosen as the coupling link between the laser-telescope part and the detection optics part, after the telescope focus, since on one side with given telescope and laser geometrical characteristics determines the DFO, and on the other side towards the PMT, all rays entering the field stop have to be collected by the PMT.

As demonstrated also from Stelmaszczyk et al. (2005) from Fig. 1a we have:

$$DFO = \frac{2 \times DTL + DT + DL}{2 \times (RFOV - TFOV + A_{tilt})} \qquad (1)$$



In case that a beam expander with an expansion factor of EX is used in the emission part of a biaxial lidar configuration, the initial laser diameter increases and the corresponding laser beam divergence decreases, by a factor of EX. Thus, the effective laser parameters (DL and TFOV) after the expansion will become respectively, $DL \times EX$ and $(TFOV \times EX)^{-1}$, in all

5 formulas. The above are approximations and holds true for ideal optical components, since in general, commercial laser beam expanders are demonstrating different efficiency regarding the expansion of the beam diameter and the reduction of the beam divergence.

With paraxial optics and small angle approximation we can extract from Fig. 1b the relation:

$$RFOV = \frac{F_{col}}{FT} \times A_{IFF} = \frac{D_{FS}}{2 \times FT}$$

(2)

10 For the use of a small bandwidth IFF with small $A_{IFF}$ it is necessary to keep the RFOV small or to increase the ratio $\frac{F_{col}}{FT}$. In biaxial lidar systems the RFOV is determined by the laser and the telescope parameters and becomes larger with shorter DFO values (Eq. 1). In addition, by increasing the $D_{FS}$ the RFOV increases, and the DFO decreases but the SNR becomes lower, since the detected lidar signal is contaminated with more light coming from the sky background.

With $A_{IFF}$ and for any $A_{tilt}$ from Fig. 1b and equations (1) and (2) we get:

$$A_{IFF} = \frac{FT}{F_{col}} \times \left[ \frac{(2 \times DTL + DT + DL)}{2DFO} + TFOV - A_{tilt} \right]$$

(3)

And thus

$$DFO = \frac{2 \times DTL + DT + DL}{2 \times \left( \frac{F_{col}}{FT} \times A_{IFF} - TFOV + A_{tilt} \right)}$$

(4)

The ratio $\frac{F_{col}}{FT}$ is limited by the diameters of $D_{col}$ and DT (compare Fig. 1b) by:

$$D_{col} \geq DT \times \frac{F_{col}}{FT} + 2 \times RFOV \times (FT + F_{col})$$

(5)

And consequently:



$$\frac{\left(\frac{Dcol}{2} - RFOV \times FT\right)}{F_{col}} \geq \frac{\left(\frac{DT}{2} + RFOV \times FT\right)}{FT} \tag{6}$$

Additional constraints are the limited diameters of the lenses, filters and beamsplitters in combination with the diameter and the focal lengths of the telescope and the collimator (i.e. FT and $F_{col}$), as well as the distance $Z_1$ which is needed to be as high as possible for mounting all the optical elements, especially for the case of multi-wavelength backscatter-Raman lidar systems. Note, that the diameter $D_{col}$ of the optical parts is limited by their rising price with diameter and decreasing

5  availability. The extreme rays in Fig. 1b, are the rays detected from the near field with laser tilt ($A_{tilt}$) and must pass through all the optics, which results in Eq. 6, here expressed for the minimum and maximum focal length of the telescope with given DFO.

$$FT_{max/min} = 0.5 \times \left(\frac{D_{col}}{2 \times RFOV} - F_{col}\right) \pm \sqrt{0.25 \times \left(F_{col} - \frac{D_{col}}{2 \times RFOV}\right)^2 - \frac{F_{col} \times DT}{2 \times RFOV}} \tag{7}$$

All these parameters must be balanced for optimum lidar performance and for a specific scientific objective. The following

10  system of equations is derived with paraxial approximation (Fig. 1b):

$$Z_1 = Z_{obj} + Z_{II}$$

$$Z_{II} = \frac{RFOV}{A_{IFF}} \times (F_{col} + FT)$$

$$\frac{D_{obj}}{2} = \frac{DT \times \frac{F_{col}}{FT}}{2} + A_{IFF} \times Z_{obj}$$

$$\xrightarrow{yields} Z_1 = \frac{1}{2 \times A_{IFF}} \times \left[D_{obj} - DT \times \frac{F_{col}}{FT} + 2 \times RFOV \times (F_{col} + FT)\right] \tag{8}$$

Following Fig. 2, the diameter of the intermediate image ($D_{II}$) formed on the eye-relief plane between the collimator and the IFF, and the diameter of the objective lens ($D_{obj}$) just behind the IFF are equal to:

$$D_{II} = \frac{DT}{FT} \times F_{col} \tag{9}$$

$$D_{obj} = D_{II} + 2 \times A_{IFF} \times Z_{obj} \tag{10}$$



The rays collected by the IFF and the objective lens, are guided through the eyepiece (lenses L3 and L4 in Fig. 1b and Fig. 2 respectively) creating the second intermediate image plane at a distance $Z_3$ behind the last surface of the eyepiece lens. With paraxial optics approximation we find from Fig. 2 the relations:

$$F_{eye} = \frac{D_{PMT}}{D_{II}} \times F_{obj} \tag{11}$$

$$D_{eye} = 2 \times \left[ A_{IFF} \times F_{obj} + \frac{\frac{D_{II}}{2} - A_{IFF} \times (Z_{obj} - F_{obj})}{F_{obj}} \times F_{eye} \right] \tag{12}$$

$$Z_3 = \left[ 1 - \frac{D_{PMT} \times (Z_{obj} - D_{obj})}{\left( F_{obj} \times \frac{DT}{FT} \times F_{col} \right)} \right] \times F_{eye} \tag{13}$$

PMTs suffer from a non-uniform spatial response of their effective surface, what may cause artifacts to lidar signals during its transduction into electrical signal. Simeonov et al., 1999, revealed that the normalized spatial uniformity on the active area of the detector varies from 0.2 up to almost 3 times the average value, defined for the central part of the detector. In order to avoid lidar signal deviations due to the spatial inhomogeneity PMT sensitivity, the detector must be placed at an

10 image of the telescopes aperture. At this place (distance $Z_3$ behind the eyepiece), the image of the lidar beam does not move with the lidar distance, and the spatial intensity distribution over the PMTs active surface does not change. In addition, an advantage of using makes the detection surface rather insensitive to several axial / radial misalignments (e.g. $\pm 4$ mm / $\pm 2$ mm) of the lens L4 and the PMT (Freudenthaler et al., 2004). However, due to difficulties in measuring the exact location of the PMT cathode with respect to the PMT housing, the alignment of the detection surface behind the L4, seems to be crucial,

15 and real ray tracing simulation revealed to be mandatory for identifying the housing accuracies.

For boundary layer measurements a low DFO height is required (see Fig. 1a), thus leading to higher values of $A_{IFF}$ (Eq. 3), larger IFF bandwidth, lower sky background suppression and finally lower SNR of the system.

Tilting the laser by an angle $A_{tilt}$ with respect to the telescope axis (Fig. 1a), allow to decrease the RFOV with constant DFO (Stelmaszczyk et al., 2005). For a given DFO we find the optimum $A_{tilt}$ by equating the maximum incident angles in the

20 telescope from infinity (i.e. $A_{tilt} + TFOV$) and from the given DFO. Thus, considering that $A_{tilt}$ is limited in the far range by the RFOV:

$$A_{tilt}^{opt} = RFOV - TFOV$$



$$(14)$$

we get through Eq. (1) and Eq. (14) the minimum DFO at optimum $A_{tilt}$ ($A_{tilt}^{opt}$):

$$\mathrm{DFO_{min}} = \frac{2 \times \mathrm{DTL} + \mathrm{DT} + \mathrm{DL}}{4 \times (\mathrm{RFOV} - \mathrm{TFOV})} \qquad (15)$$

and with $A_{tilt}^{opt}$ according to Eq. (3):

$$A_{\mathrm{IFF}} = \frac{\mathrm{FT}}{\mathrm{F_{col}}} \times \left[ \frac{2 \times \mathrm{DTL} + \mathrm{DT} + \mathrm{DL}}{4 \times \mathrm{DFO}} + \mathrm{TFOV} \right] \qquad (16)$$

The IFF allows for incident angles lower than $A_{\mathrm{IFF}}^{max}$. The smaller the filter bandwidth, the smaller is $A_{\mathrm{IFF}}^{max}$ (a filter with
bandwidth BW = 0.5 nm is leading to $A_{\mathrm{IFF}}^{max}$ = 2.9º). The extreme incident angles in the telescope (RFOV) and at the IFF
($A_{\mathrm{IFF}}$) increase with decreasing lidar range (DFO) according to Eq. (3) and Eq. (16).

In Fig.3 the variation of the maximum angle of incident rays on the IFF ($A_{\mathrm{IFF}}$) for different DFO values is presented,
regarding zero degrees and optimum laser tilt ($A_{tilt}^{opt}$), according to equations (3) and (12). The values used for the
calculations (e.g. FT, $F_{col}$, DTL, DT, DL) are provided in the following paragraph (Section 3). The maximum angle of
incident rays ($A_{\mathrm{IFF}}$) on the IFF, is decreased in about 40 % (from 1.96º to 1.15º) with an optimum laser tilt for the same DFO
(182.11 m). The two blue lines are indicating the $A_{\mathrm{IFF}}$ angles for two IFFs with BW 0.5 nm and 0.15 nm respectively (see
appendix).

**3 Evaluation of paraxial approximation with ray tracing simulations**

For evaluating the formulation presented in this study, ray tracing simulations with ZEMAX software (www.zemax.com)
have been performed. Considering that unlike ZEMAX, various aberration effects are not taken into account with thin lens
approximation, in this section is investigated how close in reality are the calculations derived in comparison with real ray
tracing simulations.

The geometrical properties of the simulated lidar system used as input parameters in paraxial approximation, are leading to a
DFO = 257 m. More precisely, a laser with DL = 8 mm  and TFOV = 0.8 mrad was considered, expanded by an ideal beam
expansion unit (EX =× 4), with an $A_{tilt}$ = 0.4 mrad. The laser beam is collected by an ideal telescope with DT = 300 mm
and FT = 600 mm, guided through a circular field stop ($D_{FS}$ = 1.5 mm) to the collimator. The distance between the



collimator and the IFF considered to be $Z_1 = 160$ mm and the near field rays incident on IFFs surface with maximum angle $A_{IFF} = 1.15^o$.

The 3D ray tracing simulations have been initiated at 532 nm, using the aforementioned values and assuming an optimized Cassegrainian telescope, collecting the image of the laser beam from far and near range. The laser beam was simulated by a

disk of source rays, which was placed at the distances of 257 m and 10000 m from the telescope and 180 mm above the optical axis. For each distance, the size of the laser disk was calculated from the TFOV. Regarding the optical components mounted after the field stop, we used lenses available from the ZEMAX database, with parameters (i.e. focal length and diameter) similar to the ones revealed from the paraxial approximation calculations.

More precisely in ZEMAX we set: the distance between the collimator and the IFF at exactly 160 mm ($Z_1$), the eyepiece at

85.64 mm after the IFF ($Z_2$), while the distance between the eyepiece and PMT at 13 mm. For the simulations we used 1′′ optics. In Fig. 4 the spot diagrams of rays from far (green spots; 10000 m) and near range (blue spots; 252 m) are demonstrated. The spot diagrams in Fig. 4a are in cosine space, demonstrating the angle with which each field point of far and near range ray is falling on the first surface of the IFF filter. Maximum field incident angle on the IFF found to be equal to 0.0196 mrad. The full field spot diagram demonstrated in Fig. 4b, refers to the surface of the PMT. As can be seen in Fig.

4b a homogeneous distribution of far and near range rays on PMT surface have been achieved, covering also the same area. The spot diameter found to be 4.6 mm, within the 5 mm diameter of effective detector aperture, revealing an overall sufficient imaging of far and near range rays, on the detector.

The relative differences between the calculated parameters from paraxial approximation and the simulations with ZEMAX are demonstrated in Table 2, and the slight discrepancies are attributed to the following reasons: (a) the slightly different

parameters of lenses used from ZEMAX database, compared to those used as input ($F_{col}$) or estimated ($F_{obj}$, $F_{eye}$) with the paraxial approximation formulation and (b) the inability of paraxial approximation to take into account all kinds of possible aberrations, in contrast to ZEMAX simulation. For example, ZEMAX simulations revealed that telescope's objective lens is focusing the near and far field rays at different planes (Fig. 5). The far field rays, are focused exactly on the focal plane of the telescope while due to the defocusing effect of the telescope, the near field rays are focused in a plane with an axial shift

of 1.4 mm on the optical axis. The mounting position of the field stop should be somewhere that both far field and near field rays are captured with the same efficiency. This position revealed from ZEMAX simulations to be ∼ 0.7 mm above the far field focus of the telescope.

## 4 Effects of possible axial misalignment on DFO

An axial shift of $Z_1$ is leading to a shift of the $A_{IFF}$, affecting the RFOV of the system, and resulting consequently to a

change of the DFO. Thus, in order to identify the contribution of a possible axial shift of the L3 optical component (presented in Fig. 1b and in Fig. 2) to the DFO, an iterative based Monte Carlo method, has been applied.





The procedure that have been followed is based on the assumption that the measured quantities like length and angle are following normal distribution. Starting from Eq. (8) and keeping the rest of the parameters constant, the iterative procedure was initiated with a $Z_1 = 160$ mm, with a varying accuracy ranging from 0.1 to 10 mm. Each value of $Z_1$ with pre-defined accuracies, is leading to a range of $A_{IFF}$ values, following also a normal distribution. From the mean value and standard deviation of each distribution the discrepancies of the DFO, have been estimated according to Eq. (4). The relative error of DFO as a result of various accuracies of $Z_1$ is demonstrated in Fig. 6 left. Assuming a system with 1'' optics, and with values regarding $FT, DT, F_{col}, Z_1$ presented already in Section 3, a mounting uncertainty of the IFF and the objective lens (L3) on the optical axis, of the order of $\pm 5$ mm will lead to DFO values with up to 3.7 % relative difference. In the right part of Fig. 6, the distribution of DFO, values is demonstrated, assuming that $Z_1 = (160 \pm 5)$ mm.

## 5 Paraxial approximation regarding field stop geometrical properties

The real performance of lidar system in terms of its optical set up and specifically for the accurate imaging of the laser beam on the focal plane of the telescope, can be controlled experimentally for example by means of a CCD camera (Engelmann et al., 2016). However, with paraxial approximation it is feasible the estimation of the diaphragms size and its location on the optical axis. The geometrical properties of the field stop depends on image field points $(x_i, y_i, z_i)$ of the laser beam (object field points, $x_b, y_b, z_b$) projected by the telescope. The latter are determined by the distance of the laser and telescope axis (DTL), by the laser divergence (TFOV) and tilt ($A_{tilt}$), and by the telescopes focal length (FT). Assuming that the laser beam is above the optical axis, transmitted to the left, the collected light travels to the right, and is projected by the telescope to the image space. In Fig. 7 this optical layout is demonstrated along with the marginal and chief rays of the object in a Cartesian system of coordinates, with starting point $(0, 0, 0)$ in the center of the telescopes primary mirror.

According to the Gaussian lens formula and treating a telescope as an ideal thin positive lens, one can get (Hecht, E., 1975):

$$\frac{1}{FT} = \frac{1}{z_b} + \frac{1}{z_i}$$

$$\xrightarrow{yields} z_i = \frac{z_b \times FT}{z_b - FT} = FT \times \frac{1}{1 - \frac{FT}{z_b}} \approx FT \times \left(1 + \frac{FT}{z_b}\right) \tag{17}$$

where $z_b$ is the distance of the object, and $z_i$ the distance of the image behind the thin lens. For near range rays the image appears to be a couple of mm further than the telescopes focal length on z optical axis, due to defocusing effect of the mirror. Assuming that the laser beam is close to the telescope axis (for small angles and paraxial approximation), and $A_{tilt}$ and DTL is valid only in y direction, then the $x_i$ (beam width) and $y_i$ (distance from optical axis) are determined in first approximation by the following equations:



$$x_i = z_i \frac{x_b}{z_b} = z_i \times \frac{DL \pm TFOV \times z_b}{z_b} = \left(\frac{DL}{z_b} \pm TFOV\right) \times z_i \tag{18}$$

$$y_i = z_i \frac{y_b}{z_b} = z_i \times \left(\frac{DTL - A_{tilt} \times z_b}{z_b}\right) = \left(\frac{DTL}{z_b} - A_{tilt}\right) \times z_i \tag{19}$$

$$y_i = 0 \text{ for } A_{tilt} = \frac{DTL}{z_b}; \ z_b = \frac{DTL}{A_{tilt}} \tag{20}$$

The distance $\left(\frac{DTL}{A_{tilt}}\right)$, is the distance at which the two central axis of the laser and the telescope will cross each other in the object space. All the equations provided above can be used for the determination of the size (height and width), and location of the field stop. The image height $y_i$ becomes zero for an object projected from infinity. The object is projected almost in the centre of the field stop on optical axis, leading the smallest area of illumination on the FOVD. As the lidar range ($z_b$) decrease and the laser beam is in near range the image height $y_i$ increases leading to larger illuminated area on the plane of the FOVD.

## 6 Summary and Conclusions

Based on thin lens approximation formulas a set of equations is derived, describing the optical design of a typical EARLINET lidar system, in this study. The limitations of a lidar optical setup are revealed through geometric optics, from the emitted laser beam to its projection on the photomultiplier. The main lidar issue studied here concerns the distance of full overlap and how this depends on the entire geometry describing the optical components in the detection unit of a lidar system, and not only on the laser-telescope geometry. The usage of IFF with small bandwidth for background suppression is limited by their small acceptance angle in near field, especially if the alignment uncertainties of the mechanical setup of the lidar optics are taken into account. Small axial shift regarding the position of the IFF, may lead up to 3.7% relative difference on the DFO. The evaluation of the paraxial approximation formulation have been done with ZEMAX ray tracing simulations, revealing an overall good performance with relative differences of the order of 5%, mainly attributed to aberrations effects that are not considered in thin lens formulation. The described formulation cannot substitute an advanced optical design software, since 3D ray tracing simulations of realistic lidar systems are necessary to reveal the necessity to use highest quality optical parts mounted with the highest possible accuracy.





**Acknowledgment**

I would like to thank Dr. Volker Freudenthaler for the helpful discussions, ideas and useful comments.

**Appendix**

The centre wavelength $\lambda o$ of an interference filter (IFF) is shifted to $\lambda s$ with an incident angle $A_{IFF}$ according to:

$$\frac{\lambda_s}{\lambda_o} = \sqrt{1 - \left[\frac{n_o}{n_e} \times \sin(A_{IFF})\right]^2} \tag{21}$$

with the effective refractive index of the filter $n_e$ and the refractive index of the environment $n_o$. The shift is to smaller wavelengths with increasing $A_{IFF}$, and the more the larger $n_e$. Examples for IFF are a Barr filter with 0.5 nm bandwidth (BW, FWHM) at 532 nm, $n_e = 1.99$ and a temperature coefficient of 0.0021 nm °C$^{-1}$, and a Andover filter with BW =

10    0.15 nm at 532 nm, $n_e = 1.45$, and temperature coefficient 0.016 nm °C$^{-1}$. The incident angles $A_{IFF}$ are limited by the maximum allowed wavelength shift for acceptable transmission, which have been set at $0.7 \times \frac{BW}{2}$, i.e. about 0.18 nm (Barr) and 0.05 nm (Andover). This results in $A_{IFF}^{max.}$ of 2.9° and 1.14°, for Barr and Andover filters respectively.

25



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



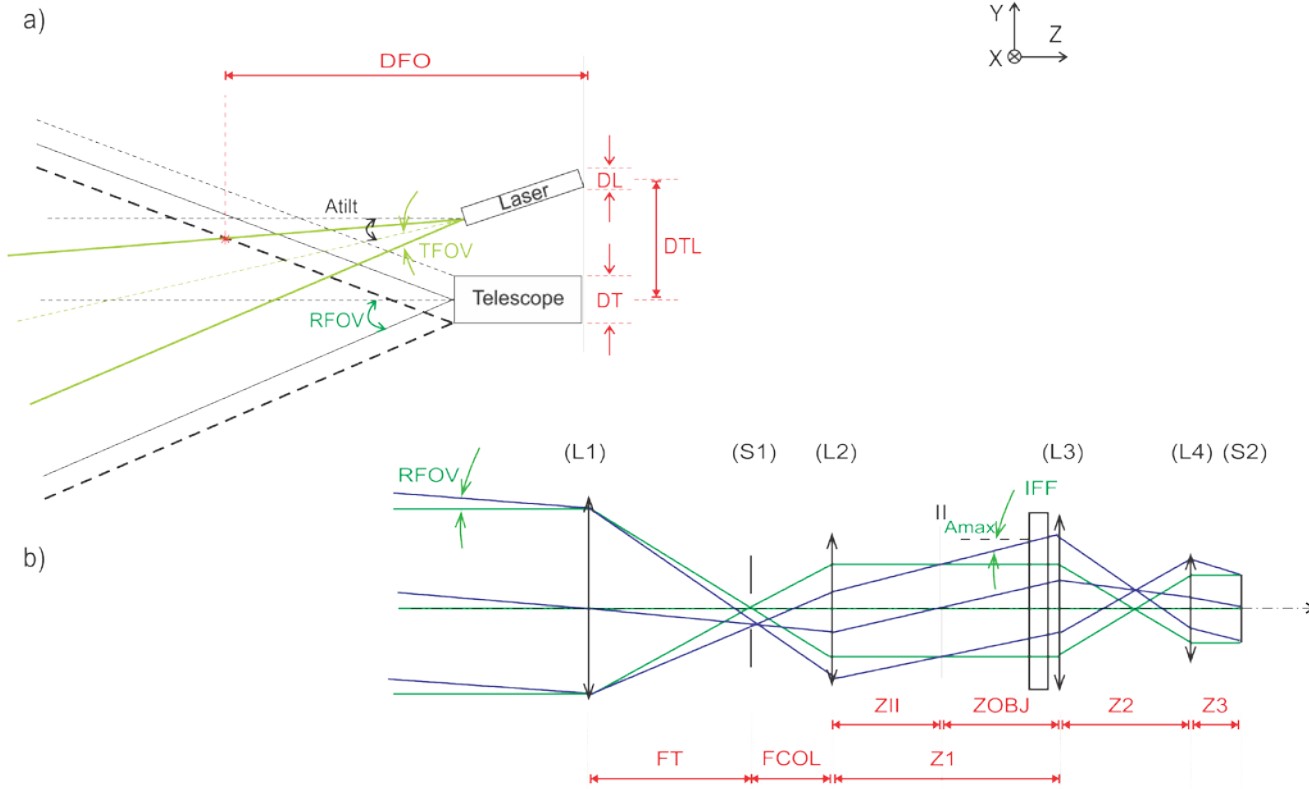

**Figure 1: (a) The laser-telescope geometry of a biaxial lidar system with a laser tilt $A_{tilt}$ and distance of full overlap DFO. RFOV and TFOV are the receiver's field of view and laser beam divergence respectively (half angles). (b) The optical setup of a lidar receiving unit with telescope (L1), field of view diaphragm FOVD (S1), collimating lens (L2), interference filter IFF and objective lens (L3). Rays collected from far (green lines) and near range (blue lines) with the maximum incident angle at the telescope (RFOV), which is limited by the FOVD, reach the IFF, with a free aperture diameter of $D_{obj}$, located at distance $Z_1$ from L1 under an incident angle $A_{IFF}$. S2 is the surface of the PMT with diameter $D_{PMT}$.**



| Object Space | |
|---|---|
| **DFO** | The distance of full overlap of the lidar system |
| **DTL** | The distance between telescope and laser central axis |
| **DT** | The clear aperture of the telescope |
| **DL** | The diameter of the laser beam |
| **FT** | The focal length of the telescope |
| **RFOV** | The receiver field of view (half angle) |
| **TFOV** | The laser beam divergence (half angle) |
| $\mathbf{A_{tilt}}$ | The inclination angle of the laser beam axis relative to the telescope axis |
| **Image Space** | |
| $\mathbf{D_{FS}}$ | The diameter of the field stop |
| $\mathbf{F_{col}}$ | The focal length of the collimating lens |
| $\mathbf{D_{col}}$ | The diameter of the collimating lens |
| $\mathbf{A_{IFF}}$ | The incidence angle of the near range rays on the interference filter |
| $\mathbf{Z_1}$ | The distance between the collimator and the objective lens |
| $\mathbf{Z_{II}}$ | The distance between the collimator and the plane of intermediate image |
| $\mathbf{D_{II}}$ | The diameter of the intermediate image |
| $\mathbf{Z_{obj}}$ | The distance between the plane of intermediate image and the objective lens |
| $\mathbf{D_{obj}}$ | The diameter of the objective lens |
| $\mathbf{Z_2}$ | The distance between the objective and the eyepiece lens |
| $\mathbf{Z_3}$ | The distance between the eyepiece lens and the detector |
| $\mathbf{F_{eye}}$ | The focal length of the eyepiece lens |
| $\mathbf{D_{eye}}$ | The diameter of the eyepiece lens |
| $\mathbf{D_{PMT}}$ | The diameter of the detector |

**Table 1: A list of the abbreviations that are used for describing the lidar parameters, along with their meaning.**





**Figure 2: The optical path of far (green) and near range (blue) rays behind the first Intermediate Image (II) plane.**




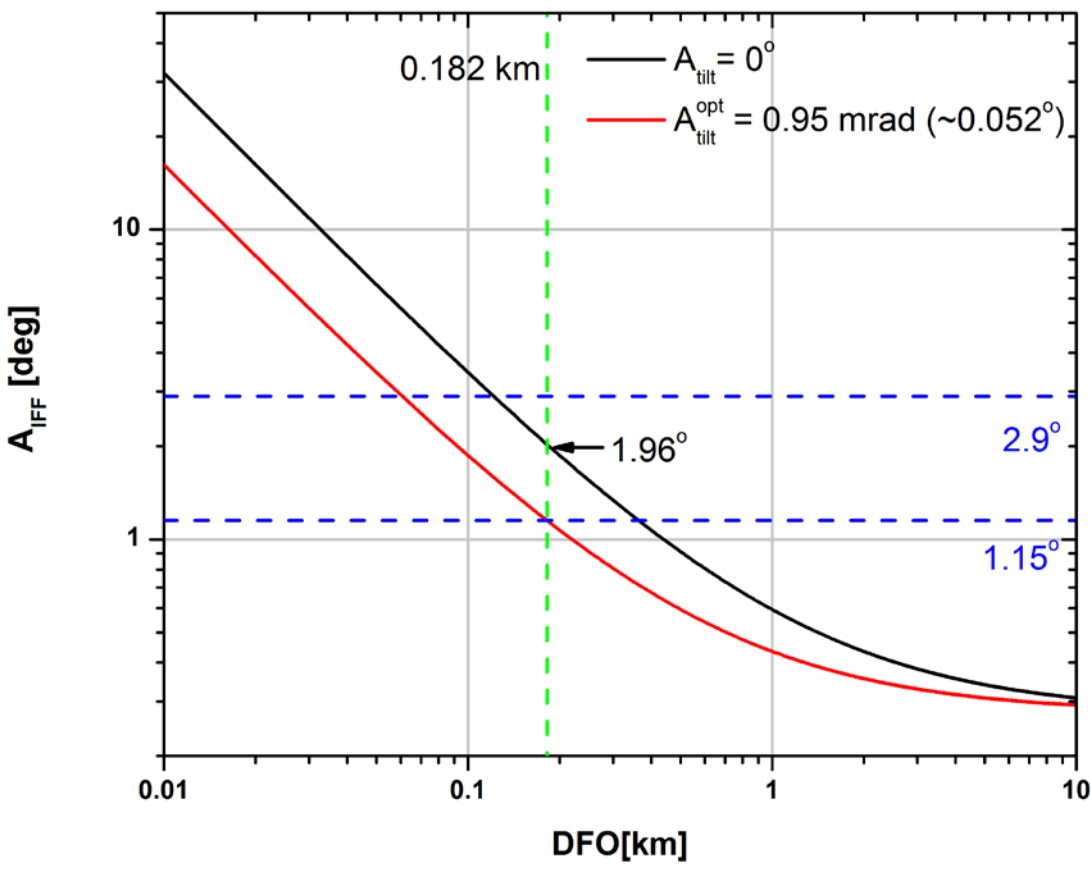

**Figure 3: The variability of the maximum angle of incident rays on the IFF ($A_{IFF}$) for different DFO values, without laser tilt ($A_{tilt} = 0^0$; black line) and with optimum laser tilt ($A_{tilt}^{opt.} = 0.95$ mrad; red line).**





**Figure 4: Spot diagrams of far (green) and near range (blue) rays on the (a) front surface of the IFF, and (b) PMT detector with 5**

5   **mm effective diameter (black circle).**





| Optical Parameter | ZEMAX | Paraxial | Relative difference (%) |
|---|---|---|---|
| $Fcol$ (mm) | 37.72 | 37.36 | 0.95 |
| $Z_1$ (mm) | 160 | 160 | 0 |
| $A_{IFF}$ (mrad) | 1.126 | 1.15 | -2.08 |
| $Z_2$ (mm) | 85.64 | 85.64 | 0 |
| $Fobj$ (mm) | 74.69 | 70.54 | 5.56 |
| $Z_3$ (mm) | 13 | 12.82 | 1.38 |
| $Feye$ (mm) | 15.31 | 15.10 | 1.42 |

**Table 2: The optical parameters (distances and focal lengths) estimated with paraxial approximation and simulated by ZEMAX, along with their relative difference.**

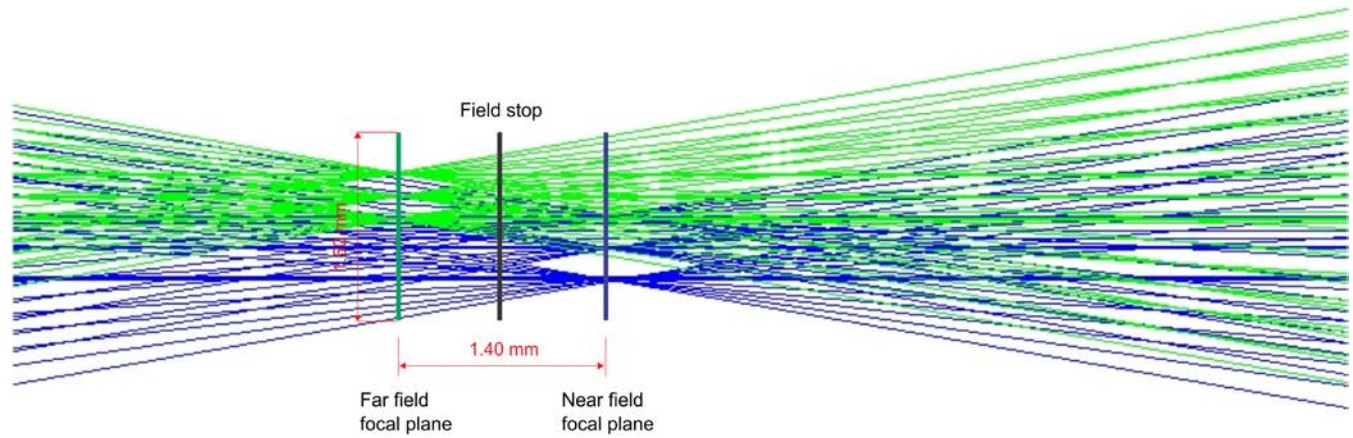

**Figure 5: The focal planes of far (green; 10000 m) and near range (blue; 257 m) rays developed behind the primary mirror of the telescope. A defocusing of 1.4 mm was estimated with ZEMAX simulations.**





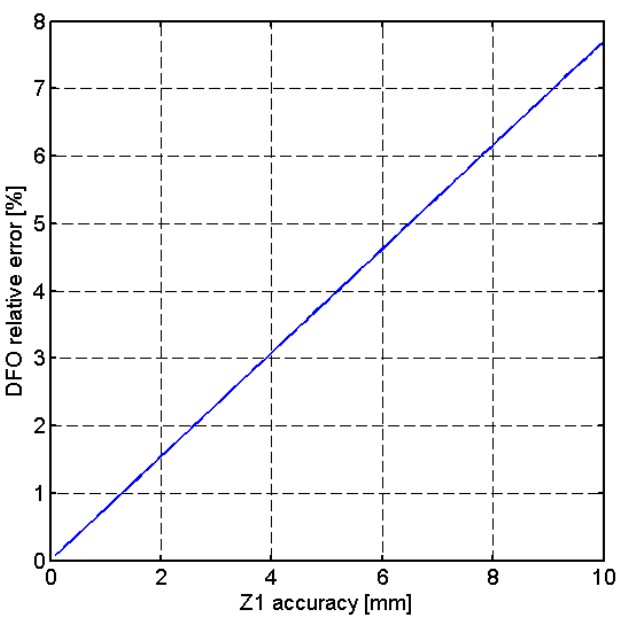
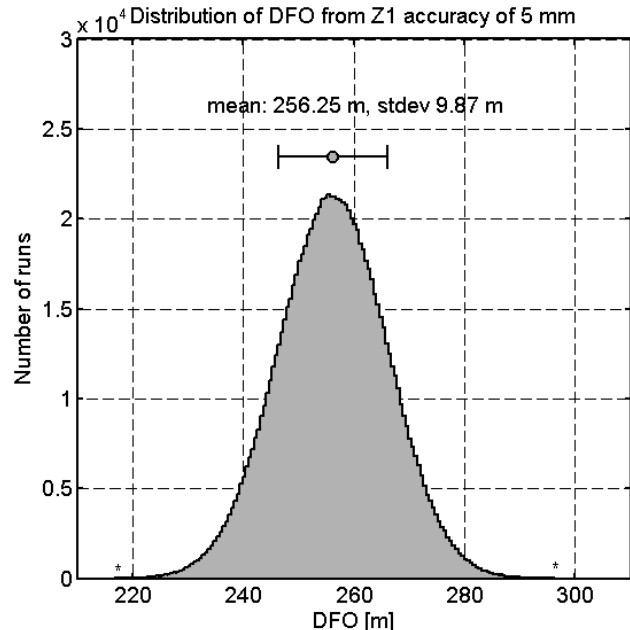

**Figure 6:** Variation of DFOrelative difference according to various $Z_1$ accuracies (left), and the DFO distribution retrieved from $10^6$ Monte Carlo runs, assuming a $Z_1$ accuracy of 5 mm (right).





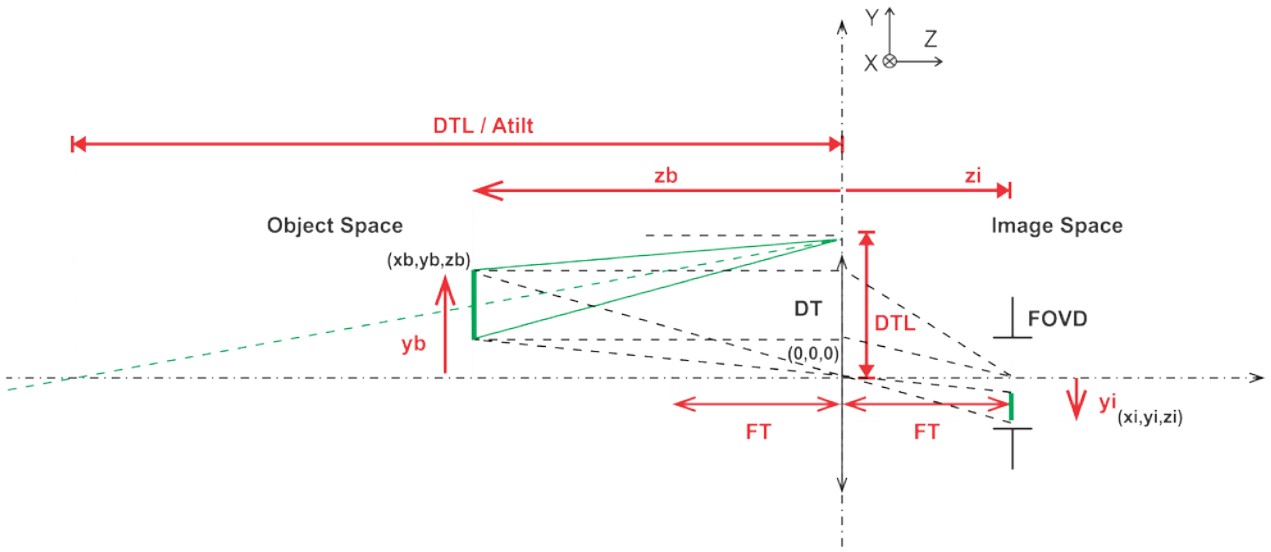

**Figure 7: The projection of laser beam field point to the FOVD through the telescope.**