# Peer review of "Using paraxial approximation to describe the optical setup of a typical EARLINET lidar system"

_Atmospheric Measurement Techniques, 2016_

## Referee Comment (RC1) · Anonymous Referee #2 · 17 Sep 2016

Although not very original in its contents, the paper has the merit of gathering in a single document the lore of good practices in lidar optical design arising from different sources and being put into effect in the EARLINET community. However, in my opinion it needs some modifications to make it more clear, emphasize some key points, clarify others and correct some mistakes. In particular more guidance for the reader through the presented formulas should be given, highlighting the important points they are exposing. Two particular points should be more emphasized in my opinion: a) the importance of the system imaging the entrance pupil of the telescope onto the surface of the photodetector, to eliminate the effect of response inhomogeneities over the surface; b) the constraint on the receiver field of view imposed by the acceptance angle of the interference filter (see specific remark No. 22b). Detailed remarks follow.

**General remarks**

1. The paper makes an extensive (perhaps excessive) use of acronyms, which makes it difficult to follow. If reducing the use of acronyms is not possible, at least a table explaining their meaning should be introduced at the beginning or the end of the paper.

2. Throughout the paper the term "primary mirror" seems to be used as synonymous of "telescope". This should be revised. While it is true that for a Newtonian telescope the focal length will be that of its primary mirror, this is not true for Cassegrainian telescopes also used in many lidar systems.

3. The approximation of the transmitted beam by a truncated cone for the purposes of the paper should be justified (see specific remark No. 4)

4. Although in general very good, the English writing should undergo a revision to polish some expressions.

**Specific remarks:**

1. Page 1, line 9: the concept of "final receiving unit" should be clarified.

2. Page 1, sentence starting on line 26: "Systematic errors are mostly linked to the estimation of temperature and pressure profiles along with the wavelength dependence parameter required in Raman technique". This seems to be too exclusive. On the one hand, interferences caused by the laser source in the analog receiving channels can be also a cause of systematic error. On the other hand, the sentence may be contradictory of some of the effects discussed in the paper (e.g. range-dependent overlap factor), which would also lead to systematic errors.

3. Page 2, sentence starting on line 10: "however without being able to provide any information related to depolarization retrievals, due to the usage of paraxial optics formalism." I don't see why the use of paraxial optics precludes dealing with depolarization retrievals. I would rather think that the paper is just not aimed at treating depolarization issues.

4. Page 2, sentence starting on line 13 about the overlap function: "it [the overlap function] describes the fraction of the laser beam cross section contained within the receiver field of

view". It should be made more precise what "contained within the receiver field of view" means. In this respect, fig. 1a is not very clear. *RFOV* is defined as the half angle of a cone with apex on the center of the telescope aperture, but a cone with apex on the lower edge is used to define the full overlap distance. Moreover, further developments seem to imply that the beam is "hard" limited by its divergence, with no energy outside it. This would seem to exclude from the treatment the widely used Gaussian beam approximation. Considerations about the approximations assumed should be included in the discussion. Probably a "generous" definition of the beam width, as the width that contains a high percentage of the energy would help in overcoming this issue.

5. Page 2, line 15: it makes me a little uneasy that the laser beam divergence is called *TFOV* (which I understand would stand for transmitter field of view). In my opinion the field of view is a parameter that is mainly used to specify receiver systems, so using it for a beam divergence may be confusing. Why not keep the term "beam divergence" and, if an acronym is necessary, just *BD*?

6. Page 2, sentence starting on line 27: "multiple scattering effects have to be taken into account especially for cases where non-spherical particles are suspended in the atmosphere (Wandinger, 1998; Wandinger et al., 2010)". I'm not sure that from the cited references it can be generally drawn that multiple scattering effects have to be taken into account *especially* in the presence of non-spherical particles. Please check if this is a general conclusion.

7. Page 3, sentence starting on line 1: "Such IFFs have recently become commercially available with small *BW* of the order of 0.17 nm (*FWHM*) at visible spectrum (Alluxa)". A more precise reference than just the manufacturer should be given for such filters.

8. Page 3, sentence starting on line 2: "A significant drawback of these filters is that a decrease bandwidth can be caused when the acceptance angle ($A_{IFF}^{max.}$) is decreased as well, which in turn limits the possible *DOF*." I find the sentence obscure; wouldn't it be the other way around, i.e, the narrow bandwidth causes a small acceptance angle? Please rephrase it to make it clearer.

9. Page 3, line 8: among the cited references related to the determination of the overlap function, the following one should be added: T. Halldórsson and J. Langerholc, "Geometrical form factors for the lidar function," *Appl. Opt.*, vol. 17, no. 2, pp. 240–244, 1978."

10. Page 4, 1st paragraph of section 2: I have a couple of issues with this paragraph: 1) on line 5 the "field of view of the telescope" is mentioned. I think the term "field of view of the receiver setup" would be more appropriate, as the field of view is determined by the telescope focal length *and* by the field stop diameter; 2) the values of the assumed focal length and field stop diameter producing the stated 1.25 mrad field of view in the example (which seems to be the same taken again in Section 3) are missing.

11. Page 4, section 2.1: the text continuously refers to fig. 1 to describe the optical layout. In this figure the layout elements are identified as L1, S1, L2, etc. These identifications should also be used in the text for the benefit of the reader.

12. Page 4, lines 22-23: it should be emphasized that the modeling of the transmitted beam by the truncated cone implied by the description is an approximation.

13. Page 4, sentence starting on line 23: "the backscattered light is collected by the primary mirror of a telescope, with a focal length *FT* and clear aperture *DT*." The sentence is somewhat ambiguous in that it is not clear if the focal length *FT* is that of the telescope or that of the primary mirror. Both focal lengths are the same for Newtonian telescopes, but not for Cassegrainian ones. It should me made clear that *FT* is the focal length of the telescope.

14. Page 4, sentence starting on line 25: "The *RFOV* (half angle) of the telescope is determined by a diaphragm *FOVD* (usually a circular iris), with diameter $D_{FS}$ centered on the optical axis, and mounted behind the primary mirror of the telescope." The field diaphragm will not only be mounted behind the primary mirror of the telescope, but also behind the secondary one. More specifically, according to fig. 1b and to Eq. (2), it is mounted on the telescope focal plane. Moreover I think this would be the place to insert the equation

$$RFOV = \frac{D_{FS}}{2 \times FT}$$ (see also remark No. 22b).

15. Page 4, line 30: the paraxial approximation assumes that rays are not too distant of the system axis and that their angles with respect to that axis are small. Chromatic aberration has instead to do with the dependence of the refractive index on wavelength. Therefore I don't think that the paraxial approximation implies neglecting chromatic aberration. Likewise, focal blur of the telescope is not necessarily associated to departures of the paraxial approximation: it appears when imaging points not sufficiently far away from the telescope, even under the paraxial approximation.

16. Page 5, sentence starting on line 1: "Initially, the rays collected by the primary mirror are coming both from far (parallel to the optical axis) and near range (with an inclination determined by the *RFOV*); (green and blue lines in Fig. 1b respectively), focused on its focal plane and thus spatially filtered by *FOVD*." I have two remarks to this sentence: 1) In my opinion, what distinguishes rays coming from points on the far or near range is not that they are parallel to the system axis or not, but rather that they are parallel between themselves or not. Rays coming from far-range points close to the field-of-view limit will be slant, yet parallel between themselves. 2) Moreover, the expression "with an inclination determined by the *RFOV*" seems to imply an action of the *RFOV* on the rays. Probably the author means that *RFOV* determines the maximum inclination of the rays (whether from far or near range) that will pass through the field diaphragm. The sentence should be rephrased to be more accurate.

17. Page 5, sentence starting on line 5: "The collimated far and near range rays are producing an intermediate image (II), the so called eye-relief plane, at a distance $Z_{II}$ behind the collimating lens". The sentence is ill-constructed, as it says that the intermediate image being produced at II is the eye-relief plane; in fact the eye-relief plane is where the image is being formed, and that image is that of the entrance pupil.

18. Page 5, sentence starting on line 9: "At this point the image projected by the telescope becomes sharper and is independent on the lidar range". Remark related to the previous one:

it should be said that this image is that of the entrance aperture. No wonder it does not depend on the lidar range, as the position of entrance aperture is fixed.

19. Page 5, line 19: "principal" should be "principle".

20. Page 5, lines 23-25. Again the first sentence seems to imply that the beam is "hard" limited in width, not accounting for, for example, Gaussian beams (see remark No. 4). I think that the second sentence "For those ranges, the extreme points of the telescope mirror and consequently each point of the telescope, detect the laser pulse entirely and with the same collecting efficiency" can be simplified by saying that, for ranges where full overlap occurs, any ray coming from a point in the illuminated volume and reaching the telescope aperture will pass through the field diaphragm. The purpose of last sentence ("This is true for small inclination angles of the laser central axis towards the telescope axis ($A_{tilt}$)" is also obscure: what would happen if $A_{tilt}$ were not small? Perhaps the overlap factor could decrease for farther ranges after reaching a peak? (see remark No. 22a)

21. Page 6, Eq. (2): the symbol $A_{IFF}$ used in this equation seems not to have been previously defined. Probably it corresponds to the $A_{IFF}^{max}$ defined on line 5 of page 5. Please check for consistence in the symbols. I'm not sure that $A_{IFF}$ is always given the same meaning (maximum acceptance angle, or angle of the rays arriving at the interference filter?).

22. Page 6, line 12: "but the SNR become lower". It should be pointed out that this will only happen in daytime operation.

23. Page 6 and ff: I find there is a lack of guidance for the reader to grasp the constraints implied by the equations. For example:

   a) I think an important constraint is missing, namely $A_{tilt} + TFOV \leq RFOV$. Otherwise, even if full overlap is reached in some range, the beam will eventually exit the full-field-of-view-zone. This, together with the condition that the denominator of Eq. (1) must be positive to have $DFO > 0$, i.e. $RFOV - TFOV + A_{tilt} > 0$, leads to the condition $RFOV - TFOV \geq 0$. Something about this is said later (Eqs. (14) to (16)), when an optimum $A_{tilt}$ is defined, but I think it should be somehow anticipated here and a warning about these design trade-offs be given.

   b) It seems to me that, in spite of the explanation below it, the meaning of Eq. (2), which is to find the limit imposed on the receiver field of view by the maximum acceptance angle of the interference filter, would be clearer if written as

   $$RFOV = \frac{D_{FS}}{2 \times FT} \leq \frac{Fcol}{FT} \times A_{IFF}$$

   c) I don't see the meaning of the logical inference implied by "And thus" between Eqs. (3) and (4). Eq. (4) can be found from (1) and (2) without the need of (3).

24. Page 6, line 11: the sentence "the $RFOV$ is determined by the laser and the telescope parameters and becomes larger with shorter $DFO$ values" sounds a little puzzling in that sense

that it seems to imply a causality relationship between *DFO* and *RFOV*, the short *DFO* being the cause of a wide *RFOV*, while it is rather the contrary: it's because the *RFOV* is large that *DFO* is small.

25.  Page 7, sentence starting on line 8: "here expressed for the minimum and maximum focal length of the telescope with given *DFO*". Does the author mean "with given *RFOV*"?

26.  The block of Eqs. (8) is difficult to understand because $Z_{II}$ is not properly defined (see remark No. 17). Moreover, to make the comprehension easier it should be said that $\dfrac{RFOV}{A_{IFF}}$ has been substituted for $\dfrac{Fcol}{F_T}$ in the $Z_{II}$ expression.

27.  Figure 2: 1) The caption should explain what the different panels are intended to demonstrate and the meaning of the different symbols (Dii, Fobj, Zobj, etc.); are all the panels really necessary?  2) The formulas on the top right of the different panels do not convey easily understandable information to the reader: they should be either explained if relevant, or removed if not. 3) What identifies rays coming for point in far or near range is not their being parallel or not to the axis, but their being parallel or not between themselves; in my opinion the green rays on the two top panels correspond rather to on-axis points and the blue ones to off-axis points.

28.  Page 8, line after Eq. (10): it is not clear what the author calls "eyepiece". In this line it seems the term designates the combination of lenses L3 and L4 ("The rays […] are guided through the eyepiece (lenses L3 and L4 in Fig. 1b and Fig. 2 respectively)"), but the used terminology, calling *Fobj* the focal length of L3 and *Feye* the focal length of L4 would lead think that what's called "eyepiece" is L4 (and L3 "objective", by the way). This should be clarified. In addition, the meanings of *Fobj* and *Feye* should be explained in the text and in the caption of fig. 2.

29.  Page 8, second line after Eq. (10): "creating the second intermediate image plane at a distance $Z_3$". It would be more precise to say "creating the final image of the entrance pupil at distance $Z_3$"; there are not more images after this one.

30.  Page 8: the paragraph starting after Eq. (13) "PMTs suffer from a non-uniform spatial response of their effective surface…" is crucial for the optical layout being described. This is the reason why one wants to create an image of the telescope aperture on the surface of the photodetector. This should be stressed and the idea be expressed earlier in the paper, perhaps appearing in the abstract and in the introduction.

31.  Page 8, sentence starting on line 10: "At this place (distance $Z_3$ behind the eyepiece), the image of the lidar beam does not move with the lidar distance, and the spatial intensity distribution over the PMTs active surface does not change". Remarks: 1) What's the "lidar beam"? Maybe the author means the "laser beam". What is meant by "lidar distance"? "Distance to the lidar" would be more precise. 2) The sentence is anyway misleading: at $Z_3$ behind the eyepiece (or lens L4) the optical system is not forming an image of the beam, but an image of the telescope aperture, therefore spreading the light coming from illuminated points in the atmosphere uniformly over the photodetector surface. Lidar systems that image

the laser beam onto the photodetector surface suffer from the inhomogeneities mentioned earlier.

32. Page 8, sentence starting on line 11: "In addition, an advantage of using makes the detection surface rather insensitive…" . Something is missing after "using".

33. Page 9, line 14: "are provided in the following paragraph (Section 3)". Remark: "are provided in Section 3" would suffice.

34. Page 10, sentence starting on line 5: "and 180 mm above the optical axis". For the sake of generality it would be better "and 180 mm from the optical axis".

35. Fig. 4: it is not clear what the 5 diagrams of fig. 4a correspond to. Do they correspond to rays coming from 10 different points, 5 in the far range and 5 in the near range, at different positions within the receiver field of view? Whatever they correspond to, it should be said both in the text and in the figure caption. Note as well that a and b are missing to identify figs. 4a and 4b.

36. Page 10, sentence starting on line 22: "For example, ZEMAX simulations revealed that telescope's primary mirror is focusing the near and far field rays at different planes (Fig. 5)". This seems to be ascribed to aberration effects ("the inability of paraxial approximation to take into account all kinds of possible aberrations, in contrast to ZEMAX simulation"). However this simply results from the paraxial formula relating object and image positions (by the way, given in the first equation of equation block (17); in fact this effect seems to be taken into account in Section 7). A simple calculation shows that, for a thin lens of 600 mm focal length, the image of a point at 10000 m from the lens will be at 600.04 mm from the lens plane, while the image of a point at 257 m will be at 601.40 mm, the difference being 1.37 mm, very close to the 1.40 mm indicated in Fig. 5. So the effect seems to be explained by paraxial optics. As to the ensuing discussion on where the field stop should be placed, in my opinion it should be dropped: the displacement of the image point being a paraxial effect, it has already been implicitly taken into account in the previous developments.

37. I don't understand Section 4. It starts by saying that "An axial shift of $Z_1$ is leading to a shift of the $A_{IFF}$", then that the shift is produced by displacing L3. But in my understanding (although not completely clear due to some possible notation inconsistences, see remark No. 21) $A_{IFF}$ is the acceptance angle of the interference filter, so I don't see how it can be affected by the shift of a lens. Even if $A_{IFF}$ refers to the angle with respect to the axis of the rays exiting L2, it cannot be changed by the displacement of an element coming after it.

38. I'm not sure section 5 is relevant, as, in my opinion, its conclusions are implicit in the considerations of the previous sections: the field stop, together with the telescope focal length, determines the receiver field of view, and all the rays reaching the telescope aperture coming from illuminated points in the full field of view zone, as defined in fig. 2 of Stelmaszczyk et al. 2005, will pass through the field stop. Moreover there are some inaccuracies:

   a) Page 11, line 20: "assuming that the primary mirror of the telescope is an ideal thin positive lens". Again this seems to assume a Newtonian telescope, which is not always the case.

   b) Page 12, line 7: "The image height $y_i$ becomes zero for an object projected from infinity." This is true for objects at finite distance from the axis, but, without further

clarifications, the sentence can contradict Eqs. (18) and (19), where, even if $z_b \to \infty$, $x_i$ and $y_i$ do not tend to 0. This is because for divergent and/or tilted beams, farther points in the beam are at farther distances of the axis (i.e. $x_b / z_b$ and $y_b / z_b$ remain constant).

39. Page 12, sentence starting on line 13: "from the emitted laser beam to its projection on the photomultiplier": this may lead to think that the laser beam is imaged onto the photomultiplier surface, which is not the case in the considered setup (see remark No. 29)

40. Page 12, sentence starting on line 16: "The usage of IFF with small bandwidth for background suppression is limited by their small acceptance angle in near field". The acceptance angle of an interference filter is independent of where the rays originate, whether in the near range or in the far range. I think that "in the near field" should be removed.

---

## Referee Comment (RC2) · Anonymous Referee #2 · 18 Sep 2016

Please note that the remarks in the detailed review refer to the initial manuscript. Some of them were already taken into account in the discussion paper under review.

---

## Referee Comment (RC3) · Anonymous Referee #1 · 20 Sep 2016

Panagiotis Kokkalis derives in his manuscript analytical formulas to calculate the overlap of a lidar system by using paraxial approximation. These are useful information for anyone who is interested in designing a lidar system, although he stated that a 3D ray tracing simulation is still necessary. The polarization effects of the optical elements cannot be described by paraxial approximation, additional literature is necessary. To assess the focal lengths and the diameters of the lenses and the acceptance angles of the interference filters, these calculations are helpful in lidar design. To study the planetary boundary layer it is desired to have a lidar system with a low distance of full overlap, which can be achieved by the presented formula. Therefore this manuscript is suitable for publication in AMT with some improvements.

[Figure]

The major remarks:

A comparison of the theoretical overlap calculations with the signals of a real EAR-LINET lidar system would be of great value. Does a lidar system with a given set of parameters (focal lengths, FOVs, . . .) reach the calculated overlap? In the current version it is only compared to ZEEMAX simulations.

The minor remarks:

- Fig 1: The DFO should start where the laser emits the beam, not at the back of the laser housing. L4 is not mentioned in the text. "with a free aperture diameter of Dobj, located at distance Z1 from L1 under an incident angle AIFF" -> it is L2 (as shown in the figure)

- Fig 2: Too much information, too less description text to explain it, not even subdivided in a, b, c. Achieve a better relation to eq. 9-13.

- Fig 3: Important figure. The acceptance angles (2.9° and 1.15°) for different BW are not explained in the description text.

- Fig 4: missing a) and b) in the picture Are the ZEEMAX calculations really from 2008? Someone may wonder, why it is such an old figure. What is fig 4a supposed to tell the reader? What exactly is shown in the 5 fields?

- Fig 6: missing space "DFO relative"

- Tab 2: indices should not be written in italic, but as normal text.

- p3, l3 "a decreased bandwidth"

- p5, l2 "Fcol"; indices should not be written in italic, but as normal text.

- p6, l3 (TFOV x EX)ˆ-1 or TFOV x EXˆ-1 ?

- p8, l12 "In addition, an advantage of using . . . makes the detection surface", something is missing
- p9, l9 "The IFF allows for incident angles lower than AIFFmax" , incomplete sentence

- p10, l11 Why do you use 252 m for the near field in the ZEEMAX calculations? The DFO is 257 m.

- p10, l19 "the slightly different parameters of lenses used from ZEMAX database" Why do you not create lenses in ZEEMAX with the same parameters as used for the paraxial approximation to exclude this source of deviations and to better assess the other effects?

- p11, l11 "The real performance of a lidar system"

- p11, l13 "However, with paraxial approximation it is feasible to estimate the diaphragms size and its location on the optical axis."

- p11, l17 "transmitted to the left, the collected light travels to the right", usually light does not travel just to the left or the right, you should make clear that you are referring to figure 7.

---

## Author Comment (AC1) · 26 Oct 2016

The author would like to thank both reviewers for their thoughtful and helpful comments and suggestions. Their reviews have made a significant contribution to the improvement of this paper. Comments of the reviewers are highlighted with grey color and answers are in normal text. Text in green color is included in the revised manuscript. The line numbering in the reviewers' comments refers to the manuscript initially submitted in Atmospheric Measurement Techniques journal, whereas the line numbering in the responses refers to the new version of the manuscript.

**REVIEWER #1:**

**General:**

Panagiotis Kokkalis derives in his manuscript analytical formulas to calculate the overlap of a lidar system by using paraxial approximation. These are useful information for anyone who is interested in designing a lidar system, although he stated that a 3D ray tracing simulation is still necessary. The polarization effects of the optical elements cannot be described by paraxial approximation, additional literature is necessary. To assess the focal lengths and the diameters of the lenses and the acceptance angles of the interference filters, these calculations are helpful in lidar design. To study the planetary boundary layer it is desired to have a lidar system with a low distance of full overlap, which can be achieved by the presented formula. Therefore this manuscript is suitable for publication in AMT with some improvements.

**The major remarks:**
A comparison of the theoretical overlap calculations with the signals of a real EARLINET lidar system would be of great value. Does a lidar system with a given set of parameters (focal lengths, FOVs,…) reach the calculated overlap? In the current version it is only compared to ZEEMAX simulations.
I would like to thank the reviewer for this comment. Indeed, in the uploaded manuscript a theoretical system has been described and simulated with ZEMAX, and finally some parameters were compared with the results from the paraxial approximation formulas.
However, the formulas described in this manuscript, in parallel with ZEMAX simulations, has been already used by the author for redesigning the multi-wavelength EARLINET Raman lidar of the National Technical University of Athens (EOLE). The system was similar to the idealized system presented in the manuscript (see Section 3), with the only exceptions that the beam expander of EOLE is x3, and the initial laser beam diameter 10 mm.
In the aforementioned values some tolerances were taken also into account (including aberration of the lenses, mounting accuracy and laser beam pointing stability ($\pm 0.05$ mrad for beam expansion x1)) leading to an effective receiver field of view equal to 1.25-0.2=1.05 mrad.
Demanding for a maximum distance of $Z_1$ ($Z_1 = 200$ mm) and $A_{IFF} = 1.5^o$, we estimated the focal lengths as well as the mounting distances of the needed lenses, with the paraxial approximation formulas. Moreover, extensive ray tracing simulations has been initiated with ZEMAX software, for optimizing the design. For the simulations we used lenses, available from ZEMAX database, with the closest specifications (focal length and diameter) to what has been already estimated with paraxial approximation. All the

aforementioned parameters are leading to a $DFO = 552$ m, while with an optimum laser tilting angle of $A_{tilt}^{opt} = 0.625$ mrad, DFO reaches the minimum value of $DFO_{min} = 276$ m.

In the framework of EARLINET network, and for keeping the highest possible standards for the operating systems, a calibration procedure is followed, every time a new system intends to become a member, or an existing member-system has been under an upgrade. For this reason, EOLE was intercompared last month with the mobile lidar system from Potenza (Italy; MUSA), which is an EARLINET reference system. The preliminary results indicated that EOLE was well aligned both in near and far range (Fig. 1a), with similar signals to be obtained from 603 m above ground (Fig. 1b).

[Figure]

**Figure 1:** (a) The normalized range corrected signals (RCS) obtained by EOLE (green line) and MUSA (red line) at 532 nm, along with their mean value (black line) versus altitude. The signals are fitted to Rayleigh signal estimated by the radiosounding data. (b) The ratio of RCS (MUSA/EOLE) is demonstrated with black solid line while the dashed black line is indicating the height range above ground in which this ratio becomes 1.

Please consider that those are preliminary results and further processing is needed for coming up also with comparison of the optical products. However, it is obvious that the simulated DFO value (552 m) and the experimental (603 m), are really close.

Moreover the entire set of equations, described in the uploaded manuscript, has been integrated in a Microsoft Excel Worksheet, through Visual Basic for Application (VBA) code, and is available to the EARLINET members, with supplementary documentation ([http://www.meteo.physik.uni-muenchen.de/~stlidar/earlinet_asos/raytracing/Basic_design/basic_lidar_design.html](http://www.meteo.physik.uni-muenchen.de/~stlidar/earlinet_asos/raytracing/Basic_design/basic_lidar_design.html)).

The aforementioned worksheet can be used as a quick check up tool of an existing lidar system.

**The minor remarks:**

- Fig 1: The DFO should start where the laser emits the beam, not at the back of the laser housing. L4 is not mentioned in the text. "with a free aperture diameter of Dobj, located at distance Z1 from L1 under an incident angle AIFF" -> it is L2 (as shown in the figure)

This is correct. I would like to thank the reviewer for pointing this mistake. Fig. 1 and its caption has been corrected according to reviewer's suggestion in the revised manuscript.

- Fig 2: Too much information, too less description text to explain it, not even subdivided in a, b, c. Achieve a better relation to eq. 9-13.

The reviewer is gratefully acknowledged for this comment.

Fig. 2 is updated including the a) b) c), for proper identification, and a better relation to Eqs. (9)-(13), has been achieved in the revised manuscript (see also the specific remark No. 27 of the second reviewer).

- Fig 3: Important figure. The acceptance angles (2.9° and 1.15°) for different BW are not explained in the description text.

The author would like to thank the reviewer for this comment. In the revised manuscript the reader is suggested to go through the appendix for the details regarding the estimation of the maximum acceptance angles of the two filters with different bandwidths. In the caption of Fig.3 the following text has been inserted:

"The blue horizontal dashed lines corresponds to the maximum acceptance angles (2.9° and 1.15°) of two IFFs with bandwidths of 0.5 and 0.15 nm respectively (see appendix)."

- Fig 4: missing a) and b) in the picture are the ZEEMAX calculations really from 2008? Someone may wonder, why it is such an old figure. What is fig 4a supposed to tell the reader? What exactly is shown in the 5 fields?

ZEMAX simulations are not for 2008, this was a mistake and I really thank the reviewer for the comment. Fig. 4 has been updated with the correct one. The missing a) and b) as well as the complete description of Fig. 4a regarding the 5 fields, has been addressed in the revised manuscript (see also answer on specific remark No. 35 of the second reviewer).

- Fig 6: missing space "DFO relative"

Corrected. Thank you.

- Tab 2: indices should not be written in italic, but as normal text.

Corrected. Thank you.

-p3, l3 "a decreased bandwidth"

Thank you for this comment. This sentence was misleading and has been changed in the revised manuscript according to your comment and the specific remark No. 8 of the second reviewer.

- p5, l2 "Fcol"; indices should not be written in italic, but as normal text.

Corrected. Thank you.

- p6, l3 (TFOV x EX)ˆ-1 or TFOV x EXˆ-1 ?

The reviewer is gratefully acknowledged for pointing this typo. In the revised manuscript this formula has been corrected to $\mathrm{LBD} \times \mathrm{EX}^{-1}$. Please note here, that the acronym for describing the laser beam divergence has been changed in the revised manuscript from $\mathrm{TFOV}$ to $\mathrm{LBD}$, according to specific remark No. 5, of the second reviewer.

- p8, l12 "In addition, an advantage of using : : : makes the detection surface", something is missing
I would like to thank both reviewers for pointing this incomplete sentence. The missing word "an eyepiece lens" has been added in the revised manuscript (see also specific remark No. 32 of the second reviewer).

- p9, l9 "The IFF allows for incident angles lower than $\mathrm{A}_{\mathrm{IFF}}^{\max}$ , incomplete sentence

Thank you once again for pointing this incomplete sentence. The entire sentence has been rephrased as follows:

"The IFF allows for acceptable transmission of the backscattered rays with incident angles lower than $\mathrm{A}_{\mathrm{IFF}}^{\max}$ (see appendix)."

- p10, l11 Why do you use 252 m for the near field in the ZEEMAX calculations? The DFO is 257 m.
The reviewer is gratefully acknowledged for pointing this typo. The DFO was simulated to be 257 m and this value has been used regarding the near field rays in ZEMAX. In the revised version of the manuscript the value of 252 m has been corrected to 257 m.

- p10, l19 "the slightly different parameters of lenses used from ZEMAX database" Why do you not create lenses in ZEEMAX with the same parameters as used for the paraxial approximation to exclude this source of deviations and to better assess the other effects?
I would like to thank the reviewer for this suggestion, but I believe that creating custom lenses in ZEMAX is out of the scope of this manuscript. However, this approach could be followed in the future.

- p11, l11 "The real performance of a lidar system"
Corrected. The word "a" has been added as suggested by the reviewer. Thank you.

- p11, l13 "However, with paraxial approximation it is feasible to estimate the diaphragms size and its location on the optical axis."
Corrected as suggested by the reviewer. Thank you.

- p11, l17 "transmitted to the left, the collected light travels to the right", usually light does not travel just to the left or the right, you should make clear that you are referring to figure 7.
Thank you. I agree with reviewer's comment. In the updated manuscript is stated that this scenario refers to Fig. 7.

---

## Author Comment (AC2) · 26 Oct 2016

The author would like to thank both reviewers for their thoughtful and helpful comments and suggestions. Their reviews have made a significant contribution to the improvement of this paper. Comments of the reviewers are highlighted with grey color and answers are in normal text. Text in green color is included in the revised manuscript. The line numbering in the reviewer's comments refers to the manuscript initially submitted in Atmospheric Measurement Techniques journal, whereas the line numbering in the responses refers to the new version of the manuscript.

**REVIEWER #2:**

**General:**

Although not very original in its contents, the paper has the merit of gathering in a single document the lore of good practices in lidar optical design arising from different sources and being put into effect in the EARLINET community. However, in my opinion it needs some modifications to make it more clear, emphasize some key points, clarify others and correct some mistakes. In particular more guidance for the reader through the presented formulas should be given, highlighting the important points they are exposing. Two particular points should be more emphasized in my opinion: a) the importance of the system imaging the entrance pupil of the telescope onto the surface of the photodetector, to eliminate the effect of response inhomogeneities over the surface; b) the constraint on the receiver field of view imposed by the acceptance angle of the interference filter (see specific remark No. 22b). Detailed remarks follow.

**General remarks:**

1. The paper makes an extensive (perhaps excessive) use of acronyms, which makes it difficult to follow. If reducing the use of acronyms is not possible, at least a table explaining their meaning should be introduced at the beginning or the end of the paper.

I agree that the many abbreviations and symbols used in this manuscript may lead to confusion. Therefore a summarizing table (Table 1) has been added to the manuscript which includes the abbreviations used for the description of the lidar system in this study, along with a brief explanation.

2. Throughout the paper the term "primary mirror" seems to be used as synonymous of "telescope". This should be revised. While it is true that for a Newtonian telescope the focal length will be that of its primary mirror, this is not true for Cassegrainian telescopes also used in many lidar systems.

The reviewer is right at that point. The entire manuscript has been corrected, by changing appropriately the term "primary mirror" with the word "telescope".

3. The approximation of the transmitted beam by a truncated cone for the purposes of the paper should be justified (see specific remark No. 4)

Thank you for this comment. In the revised manuscript this is now justified (see answer on specific remark No. 4).

4. Although in general very good, the English writing should undergo a revision to polish some expressions.

I tried to go over the manuscript several times to smoothen the text. However, I would like to apologize in case that some expressions are not polished sufficiently or some grammar mistakes are still left in the manuscript.

**Specific remarks:**

1. Page 1, line 9: the concept of "final receiving unit" should be clarified.

Following the reviewer's suggestion, I clarified the final receiving unit concept in the revised manuscript (page 1, line 2) as follows:

"The equations describing a lidar system from the emitted laser beam to the projection of the telescope aperture on the final receiving unit (i.e. photomultiplier) are revealed, based on paraxial approximation and geometric optics approach."

2. Page 1, sentence starting on line 26: "Systematic errors are mostly linked to the estimation of temperature and pressure profiles along with the wavelength dependence parameter required in Raman technique". This seems to be too exclusive. On the one hand, interferences caused by the laser source in the analog receiving channels can be also a cause of systematic error. On the other hand, the sentence may be contradictory of some of the effects discussed in the paper (e.g. range-dependent overlap factor), which would also lead to systematic errors.

I would like to thank the reviewer for this comment. Indeed, what was mentioned regarding the sources of systematic error affecting the lidar retrievals was too exclusive. In the revised manuscript (page 2, line 1) I tried to be a little bit more generic, rephrasing this sentence as follows:

"On the other hand, systematic errors may arise both from assumptions or uncertain values that entering the lidar data analysis, and from the system setup and geometry. The first category may include uncertainties introduced by the estimation of temperature and pressure profiles along with the wavelength dependence parameter required in Raman technique (Ansmann et al., 1990; Whiteman, 1999), and the assumption of lidar ratio, reference height and backscattering ratio, required in backscatter technique. The second broad category of systematic errors, may include uncertainties introduced by e.g. interferences caused by the laser source in the analog receiving channels, the range-depended overlap factor, the calibration of the system etc."

3. Page 2, sentence starting on line 10: "however without being able to provide any information related to depolarization retrievals, due to the usage of paraxial optics formalism." I don't see why the use of paraxial optics precludes dealing with depolarization retrievals. I would rather think that the paper is just not aimed at treating depolarization issues.

Following the reviewer's suggestion, this text has been rephrased in the revised manuscript (page 2, line 16) as follows:

"The current study is complementary to the aforementioned work, since it is also linked with the detected lidar signal, through the usage of paraxial optics formalism, however without aiming to the treatment of depolarization issues."

4. Page 2, sentence starting on line 13 about the overlap function: "it [the overlap function] describes the fraction of the laser beam cross section contained within the receiver field of view". It should be made more precise what "contained within the receiver field of view" means. In this respect, fig. 1a is not very clear. RFOV is defined as the half angle of a cone with apex on the center of the telescope aperture, but a cone with apex on the lower edge is used to define the full overlap distance. Moreover, further developments seem to imply that the beam is "hard" limited by its divergence, with no energy outside it. This would seem to exclude from the treatment the widely used Gaussian beam approximation. Considerations about the approximations assumed should be included in the discussion. Probably a "generous" definition of the beam width, as the width that contains a high percentage of the energy would help in overcoming this issue.

I would like to thank the reviewer for this comment. In the revised manuscript I tried to make a clearer statement regarding the definition of the distance of full overlap, and relate it properly with Fig. 1a., by inserting the following text:

"The lidar equation in its simplest form includes the overlap function ($O(z)$) and the overall optical efficiency of the system. The overlap function is range-dependent and thus related to the lidar system geometry, since it describes the fraction of the light scattered within the receiver field of view, taking values from 0 to 1 (Wandinger, 2005). More precisely, at the height range where the overlap function reaches the value of 1, the lower edge (extreme) points of the telescope and consequently each point of it, collects the scattered light entirely and with the same efficiency (Fig. 1a). This height range is determined by the intersection point between the outer edge of the laser beam divergence (LBD) and the lateral surface of receiver field of view (RFOV) cone, with apex the lower point of the telescope (Fig. 1a). "

The reviewer's comment for taking into consideration the energy of a Gaussian laser beam, is discussed in Section 2.1 of the revised manuscript (page 5, line2). At that point, the approximation followed for modeling the transmitted laser beam in the atmosphere is presented (see answer to specific remark No. 12).

5. Page 2, line 15: it makes me a little uneasy that the laser beam divergence is called TFOV (which I understand would stand for transmitter field of view). In my opinion the field of view is a parameter that is mainly used to specify receiver systems, so using it for a beam divergence may be confusing. Why not keep the term "beam divergence" and, if an acronym is necessary, just BD?

I agree with the reviewer that the acronym TFOV, for describing the laser beam divergence is misleading. Thus, the acronym TFOV has been replaced with the acronym "LBD" (stands for laser beam divergence), in the text, formulas and figures (Fig. 1), of the revised manuscript.

6. Page 2, sentence starting on line 27: "multiple scattering effects have to be taken into account especially for cases where non-spherical particles are suspended in the atmosphere (Wandinger, 1998; Wandinger et al., 2010)". I'm not sure that from the cited references it can be generally drawn that multiple scattering effects have to be taken into account especially in the presence of non-spherical particles. Please check if this is a general conclusion.

The reviewer is correct at this point. In general, the multiple scattering effect have to be taken into account when lidar systems with wide RFOV are employed, specifically for studying optically thick targets. This is now clarified in the revised manuscript as follows:

"multiple scattering effects have to be taken into account especially for case studies of optically thick targets (i.e. water and ice clouds) (e.g. Eloranta, 1998; Wandinger, 1998)"

Moreover, the following reference has been added to the Reference section of the revised manuscript.

"Eloranta, E. W.: Practical model for the calculation of multiply scattered lidar returns, Appl. Opt., 37(12), 2464, doi:10.1364/AO.37.002464, 1998."

7. Page 3, sentence starting on line 1: "Such IFFs have recently become commercially available with small BW of the order of 0.17 nm (FWHM) at visible spectrum (Alluxa)". A more precise reference than just the manufacturer should be given for such filters.

Following the reviewer's suggestion, this sentence has been revised as follows:

"Such IFFs have recently become commercially available with small BW of the order of 0.17 nm (FWHM) at visible spectrum  and high transmission values (greater than 90%) at peak (Alluxa, CA, http://www.alluxa.com). Their high transmission and narrow BW characteristics have been used recently for rotational Raman measurements at visible (Veselovskii et al., 2015) and infrared spectrum (Haarig et al., 2016)."

The following references have been added in reference section of the revised manuscript.

"Veselovskii, I., Whiteman, D. N., Korenskiy, M., Suvorina, A. and Pérez-Ramírez, D.: Use of rotational Raman measurements in multiwavelength aerosol lidar for evaluation of particle backscattering and extinction, Atmospheric Meas. Tech., 8(10), 4111–4122, doi:10.5194/amt-8-4111-2015, 2015."

"Haarig, M., Engelmann, R., Ansmann, A., Veselovskii, I., Whiteman, D. N. and Althausen, D.: 1064 nm rotational Raman lidar for particle extinction and lidar-ratio profiling: cirrus case study, Atmospheric Meas. Tech., 9(9), 4269–4278, doi:10.5194/amt-9-4269-2016, 2016."

8. Page 3, sentence starting on line 2: "A significant drawback of these filters is that a decrease bandwidth can be caused when the acceptance angle ($A_{IFF}^{max}$) is decreased as well, which in turn limits the possible DOF." I find the sentence obscure; wouldn't it be the other way around, i.e, the narrow bandwidth causes a small acceptance angle? Please rephrase it to make it clearer.

I agree with the reviewer that this sentence was obscure. Thank you. In the revised manuscript this sentence has been rephrased to:

"A significant drawback of these filters is that their narrow bandwidth can cause low acceptance angle ($A_{IFF}^{max.}$), which in turn limits the possible DFO."

9. Page 3, line 8: among the cited references related to the determination of the overlap function, the following one should be added: T. Halldórsson and J. Langerholc, "Geometrical form factors for the lidar function," Appl. Opt., vol. 17, no. 2, pp. 240–244, 1978."

The suggested reference has been added in the text and the reference section of the revised manuscript.

"Halldórsson, T. and Langerholc, J.: Geometrical form factors for the lidar function, Appl. Opt., 17(2), 240, doi:10.1364/AO.17.000240, 1978."

10. Page 4, 1st paragraph of section 2: I have a couple of issues with this paragraph: 1) on line 5 the "field of view of the telescope" is mentioned. I think the term "field of view of the receiver setup" would be more appropriate, as the field of view is determined by the telescope focal length and by the field stop diameter; 2) the values of the assumed focal length and field stop diameter producing the stated 1.25 mrad field of view in the example (which seems to be the same taken again in Section 3) are missing.

Thank you for these comments. Following the reviewer's suggestion the text "field of view of the telescope" has been changed to "field of view of the receiver setup". More precisely, the entire sentence has been rephrased to:

"On the other hand, the angular magnification is increased by 60, which means that 1.25 mrad field of view of the receiver setup (i.e. determined by telescope with focal length 600 mm and a field stop with diameter 1.5 mm), is magnified to about 75 mrad (~ 4.3°)."

11. Page 4, section 2.1: the text continuously refers to fig. 1 to describe the optical layout. In this figure the layout elements are identified as L1, S1, L2, etc. These identifications should also be used in the text for the benefit of the reader.

Following the reviewer's suggestion, the layout elements of L1, S1, L2 etc. (as identified in Fig. 1 b), have been inserted in the text of the revised manuscript. Thank you.

12. Page 4, lines 22-23: it should be emphasized that the modeling of the transmitted beam by the truncated cone implied by the description is an approximation.

Thank you for this comment. In the revised manuscript, the approximation used for modelling the transmitted beam has been emphasized as follows:

"The modeling of a transmitted laser beam in the atmosphere has been approximated by a truncated cone of an ideally circularly shaped beam with initial diameter DL and divergence LBD (half angle). The DL and LBD values provided by the manufacturers, usually correspond to the 86.5 % (2σ) of the Gaussian beam energy. In lidar optical systems, the highest possible of the laser energy is needed, and accounting for a Gaussian laser beam containing the 98.9 % (3σ) of the beam energy, both DL and LBD have to be reduced by a factor of 0.5. "

13. Page 4, sentence starting on line 23: "the backscattered light is collected by the primary mirror of a telescope, with a focal length FT and clear aperture DT." The sentence is somewhat ambiguous in that it is not clear if the focal length FT is that of the telescope or that of the primary mirror. Both focal lengths are the same for Newtonian telescopes, but not for Cassegrainian ones. It should me made clear that FT is the focal length of the telescope.

The author would like to thank the reviewer for this comment. He corrected this by clearly stating that FT is the focal length of the telescope.

14. Page 4, sentence starting on line 25: "The RFOV (half angle) of the telescope is determined by a diaphragm FOVD (usually a circular iris), with diameter $D_{FS}$ centered on the optical axis, and mounted behind the primary mirror of the telescope." The field diaphragm will not only be mounted behind the primary mirror of the telescope, but also behind the secondary one. More specifically, according to fig. 1b and to Eq. (2), it is mounted on the telescope focal plane. Moreover I think this would be the place to insert the equation $RFOV = \frac{D_{FS}}{2 \times FT}$

The reviewer is gratefully acknowledged for this comment. I agree that this phrase was misleading and has been corrected in the revised manuscript (page 5, line 7) as follows:

"The focal length of the telescope and the diaphragm FOVD are determining the RFOV (half angle) of the receiver setup, according to $RFOV = \frac{D_{FS}}{2 \times FT}$. The diaphragm is usually a circular iris, with diameter $D_{FS}$ centered on the optical axis, and mounted on telescope's focal plane."

15. Page 4, line 30: the paraxial approximation assumes that rays are not too distant of the system axis and that their angles with respect to that axis are small. Chromatic aberration has instead to do with the dependence of the refractive index on wavelength. Therefore I don't think that the paraxial approximation implies neglecting chromatic aberration. Likewise, focal blur of the telescope is not necessarily associated to departures of the paraxial approximation: it appears when imaging points not sufficiently far away from the telescope, even under the paraxial approximation.

The reviewer is gratefully acknowledged for this comment. In the revised manuscript the chromatic aberration and the focal blur of the telescope have been deleted from these lines.

16. Page 5, sentence starting on line 1: "Initially, the rays collected by the primary mirror are coming both from far (parallel to the optical axis) and near range (with an inclination determined by the RFOV); (green and blue lines in Fig. 1b respectively), focused on its focal plane and thus spatially filtered by FOVD." I have two remarks to this sentence:

1) In my opinion, what distinguishes rays coming from points on the far or near range is not that they are parallel to the system axis or not, but rather that they are parallel between themselves or not. Rays coming from far-range points close to the field-of-view limit will be slant, yet parallel between themselves.

2) Moreover, the expression "with an inclination determined by the RFOV" seems to imply an action of the RFOV on the rays. Probably the author means that RFOV determines the maximum inclination of the rays (whether from far or near range) that will pass through the field diaphragm. The sentence should be rephrased to be more accurate.

I would like to thank the reviewer for pointing this misleading issue.

Near range rays are produced by off-axis points of the object, and these rays in general are not parallel between themselves, due to finite LBD, Atilt and DTL values. However, in this study all the rays are considered to be parallel in a first approximation.

"Initially, the rays coming both from far and near range are collected by the telescope. Only the rays reaching telescope with maximum incident angle of RFOV will be collected, focused on telescope's focal plane and thus spatially filtered by FOVD (S1). Any rays incident at the telescope with angles higher than RFOV, will not pass through the diaphragm. In a typical biaxial lidar, the near range rays will not be

parallel between themselves and arrive with high incidence angle, on the edge of the RFOV. In a first approximation, blue lines in Fig. 1b may be considered to come from near range while green lines from far range."

The caption of Fig. 1 has been revised as follows:

"Figure 1: (a) The laser-telescope geometry of a biaxial lidar system with a laser tilt $A_{tilt}$ and distance of full overlap DFO. RFOV and LBD are the receiver's field of view and laser beam divergence respectively (half angles). (b) The optical setup of a lidar receiving unit with telescope (L1), field of view diaphragm FOVD (S1), collimating lens (L2), interference filter IFF and objective lens (L3), and an eyepiece lens (L4). Rays collected from on-axis (green lines) and off-axis points (blue lines) with the maximum incident angle at the telescope (RFOV), which is limited by the FOVD, reach the IFF, with a free aperture diameter of $D_{obj}$, located at distance $Z_1$ from L2 under an incident angle $A_{IFF}$. S2 is the surface of the PMT with diameter $D_{PMT}$."

17. Page 5, sentence starting on line 5: "The collimated far and near range rays are producing an intermediate image (II), the so called eye-relief plane, at a distance $Z_{II}$ behind the collimating lens". The sentence is ill-constructed, as it says that the intermediate image being produced at II is the eye-relief plane; in fact the eye-relief plane is where the image is being formed, and that image is that of the entrance pupil.

I would like to thank the reviewer for pointing this ill-constructed sentence. This sentence has been revised as follows:

"The collimating lens (L2) produces an intermediate image (II) of the entrance pupil, at a distance $Z_{II}$ behind that lens. More precisely, the intermediate image is formed at the so called eye-relief plane, where the far and near range rays are crossing each other. In case that an optical detection device is mounted there, it is feasible to obtain the full viewing angle."

18. Page 5, sentence starting on line 9: "At this point the image projected by the telescope becomes sharper and is independent on the lidar range". Remark related to the previous one: it should be said that this image is that of the entrance aperture. No wonder it does not depend on the lidar range, as the position of entrance aperture is fixed.

I would like to thank the reviewer for pointing out this. In the revised manuscript this sentence has been changed to the following:

"At this plane the image of the entrance aperture is projected by the telescope and becomes sharper."
The statement implying that the image of the entrance aperture is independent on the lidar range has been deleted.

19. Page 5, line 19: "principal" should be "principle".
Done. Thank you.

20. Page 5, lines 23-25. Again the first sentence seems to imply that the beam is "hard" limited in width, not accounting for, for example, Gaussian beams (see remark No. 4). I think that the second sentence "For those ranges, the extreme points of the telescope mirror and consequently each point of the telescope, detect the laser pulse entirely and with the same collecting efficiency" can be simplified by saying that, for ranges where full overlap occurs, any ray coming from a point in the illuminated volume and reaching the telescope aperture will pass through the field diaphragm. The purpose of last sentence ("This is true for small inclination angles of the laser central axis towards the telescope axis (Atilt)" is also obscure: what would happen if Atilt were not small? Perhaps the overlap factor could decrease for farther ranges after reaching a peak? (see remark No. 22a)

I would like to thank the reviewer for this comment. Following the reviewer's suggestion, this sentence has been simplified in the revised manuscript as follows:

"For those ranges, any ray coming from a point in the illuminated volume and is reaching the telescope aperture will pass through the field stop diaphragm."

Indeed the last sentence of this paragraph is obscure at this point and probably confusing. Moreover, since the constraints introduced by $A_{tilt}$ are also discussed later (see specific remark No. 23a), this sentence has been deleted.

21. Page 6, Eq. (2): the symbol $A_{IFF}$ used in this equation seems not to have been previously defined. Probably it corresponds to the $A_{IFF}^{max}$ defined on line 5 of page 5. Please check for consistence in the symbols. I'm not sure that is always given the same meaning (maximum acceptance angle, or angle of the rays arriving at the interference filter?).

I would like to thank you for this comment. Indeed, $A_{IFF}$ corresponds to the incidence angle of rays arriving at the interference filter, and $A_{IFF}^{max}$ corresponds to the maximum acceptance angle of the interference filter. The manuscript has been revised for consistency in the symbols and changes are made through out where appropriate. Thank you.

22. Page 6, line 12: "but the SNR become lower". It should be pointed out that this will only happen in daytime operation.

Correct. Thank you. This is now clarified in the revised manuscript through the following text:

"but the SNR becomes lower, especially during daytime conditions, where the detected lidar signal is contaminated with more light coming from the sky background."

23. Page 6 and ff: I find there is a lack of guidance for the reader to grasp the constraints implied by the equations. For example:

a) I think an important constraint is missing, namely. Otherwise, even if full overlap is reached in some range, the beam will eventually exit the full- field-of-view-zone. This, together with the condition that the denominator of Eq. (1) must be positive to have $DFO > 0$, i.e. $RFOV - TFOV + A_{tilt} > 0$, leads to the condition $RFOV - TFOV > 0$. Something about this is said later (Eqs. (14) to (16)), when an optimum $A_{tilt}$ is defined, but I think it should be somehow anticipated here and a warning about these design trade-offs be given.

The reviewer is gratefully acknowledged for pointing out this issue. Following his suggestion this important constraint is discussed at this point of the revised manuscript (page 6, line 20):

"Even if the full overlap is reached in some range, the laser beam will eventually exit the full field of view zone in the far range. This, together with the condition that the denominator of Eq. (1) must be positive to have $\mathrm{DFO} > 0$, leads to $\mathrm{RFOV} - \mathrm{TFOV} + A_{\mathrm{tilt}} > 0$. In case that the inclination angle exceeds the difference between the receiver's field-of-view and the laser's divergence (half angles), the laser beam will eventually exit the full field of view zone. The further increase of the tilting angle will result in the faster exit of the laser beam from the full field of view zone, introducing a decrease of the overlap function for farther ranges after reaching a peak."

b) It seems to me that, in spite of the explanation below it, the meaning of Eq. (2), which is to find the limit imposed on the receiver field of view by the maximum acceptance angle of the interference filter, would be clearer if written as

$$\mathrm{RFOV} = \frac{D_{\mathrm{FS}}}{2 \times \mathrm{FT}} \leq \frac{F_{\mathrm{col}}}{\mathrm{FT}} \times A_{\mathrm{IFF}}$$

Correct. Thank you. Eq. 2 has been modified according to reviewer's suggestion.

c) I don't see the meaning of the logical inference implied by "And thus" between Eqs. (3) and (4). Eq. (4) can be found from (1) and (2) without the need of (3).

Correct. The phrase "And thus" had no logical inference and was misleading. Instead, in the revised manuscript the following text has been inserted:

"From Eqs. (1) and (2) ensues that"

24. Page 6, line 11: the sentence "the RFOV is determined by the laser and the telescope parameters and becomes larger with shorter DFO values" sounds a little puzzling in that sense that it seems to imply a causality relationship between DFO and RFOV, the short DFO being the cause of a wide RFOV, while it is rather the contrary: it's because the RFOV is large that DFO is small.

Thank you. Indeed the reviewer was correct that this phrase was misleading. In the revised manuscript this sentence has been changed to:

"the RFOV is determined by the parameters of the receiver setup (FT and $D_{\mathrm{FS}}$), and the higher the RFOV, the lower the DFO ranges that may be achieved (Eq. 1)."

25. Page 7, sentence starting on line 8: "here expressed for the minimum and maximum focal length of the telescope with given DFO". Does the author mean "with given RFOV"?

Correct. Thank you. This typo has been corrected as pointed out by the reviewer.

26. The block of Eqs. (8) is difficult to understand because $Z_{\mathrm{II}}$ is not properly defined (see remark No. 17). Moreover, to make the comprehension easier it should be said that $\frac{\mathrm{RFOV}}{A_{\mathrm{IFF}}}$ has been substituted for $\frac{F_{\mathrm{col}}}{\mathrm{FT}}$ in the $Z_{\mathrm{II}}$ expression.

Thank you. The parameter of $Z_{\mathrm{II}}$ has been properly defined as suggested by the reviewer in his No. 17 specific remark. Following the reviewer's suggestion I inserted the following text in the revised manuscript:

"Furthermore, in the $Z_{II}$ expression that appears in the block of Eqs. (8), the term $\frac{RFOV}{A_{IFF}}$ has been substituted for $\frac{F_{col}}{FT}$."

27. Figure 2: 1) The caption should explain what the different panels are intended to demonstrate and the meaning of the different symbols (Dii, Fobj, Zobj, etc.); are all the panels really necessary? 2) The formulas on the top right of the different panels do not convey easily understandable information to the reader: they should be either explained if relevant, or removed if not. 3) What identifies rays coming for point in far or near range is not their being parallel or not to the axis, but their being parallel or not between themselves; in my opinion the green rays on the two top panels correspond rather to on-axis points and the blue ones to off-axis points.

Both reviewers are gratefully acknowledged for pointing these issues regarding Fig. 2 (see also the second comment of the first reviewer).

1) In the revised manuscript the characters a) b) and c) has been added in Fig. 2 and its caption has been changed to: "Figure 2: Optical path of the on-axis (green line) and off-axis (blue line) points, of the first intermediate image (II) of the telescope aperture with diameter $D_{II}$ formed at distance $Z_{obj}$ before the objective lens (L3). Through the objective lens (L3) with focal length $F_{obj}$ and an eyepiece lens (L4) with focal length $F_{eye}$, the intermediate image is formed again on the surface of the photodetector (S2), at a distance $Z_3$ behind the eyepiece lens. With green dashed line in (c) the chief ray of the $D_{II}$ is denoted. For reasons of simplicity the IFF is not included in this figure. The focal planes of each lens are denoted with vertical red lines on the principal axis."

The two upper panels possibly could have been merged, however I considered to keep them separated, for reasons of clarity and for helping the reader to go analytically through the geometry for deriving the Eqs. (9)-(13).

2) I would like to thank the reviewer for this useful comment. Maybe it's a duplication, since these formulas are found also in the manuscript, however I consider that they are providing useful information to the reader for understanding easily the geometric approach followed for the derivation of Eqs. (9)-(13). Moreover, a short description regarding these formulas, is given in the text of the revised manuscript.

"More precisely, the intermediate image of the telescope aperture (with diameter $D_{II}$) is formed initially at distance $Z_{obj}$ before the objective lens (L3) (Fig. 1b and Fig. 2). In the setup presented in Fig. 2, the intermediate image is now the object with diameter $D_{II}$ that has to be projected on the surface of the photodetector (S2), through the system of two lenses. An objective lens (L3), with focal length $F_{obj}$, and an eyepiece lens (L4), with focal length $F_{eye}$. The total magnification of that lens system is $M_{3,4} = \frac{F_{obj}}{F_{eye}}$.

Therefore, the final image of the object ($D_{II}$) projected on (S2) would have a diameter (Fig. 2a):

$$D_{PMT} = M_{3,4} \times D_{II} \rightarrow F_{eye} = \frac{D_{PMT}}{D_{II}} \times F_{obj}$$

The objective lens (L3) will form an image of an off-axis point of the object (diameter $R_1$) with a diameter $R_3 = \frac{R_1}{F_{obj}} \times F_{eye}$ located at a distance of $R_2 = A_{IFF} \times F_{obj}$ above the principal axis (Fig. 2b).

The free aperture of the lens (L4) in order to collect both off and on-axis points of $D_{II}$, has to be $D_{eye}$ (Fig. 2b):

$$D_{eye} = 2 \times \left[ A_{IFF} \times F_{obj} + \frac{\frac{D_{II}}{2} - A_{IFF} \times (Z_{obj} - F_{obj})}{F_{obj}} \times F_{eye} \right]$$

The image size of an object with diameter $R_{ii} = \frac{D_{ii}}{2}$, which is formed by lens L3 on its focal plane will be $R_4 = F_{obj} \times \frac{\frac{D_{II}}{2}}{Z_{obj} - F_{obj}}$ (Fig. 2c). Subsequently, $R_4$ is the size of an object that will form an image through the eyepiece lens (L4), on the surface of the photodetector (S2) mounted at distance $Z_3$ (Fig. 2c), equal to:"

$$Z_3 = \left[ 1 - \frac{D_{PMT} \times (Z_{obj} - D_{obj})}{\left( F_{obj} \times \frac{DT}{FT} \times F_{col} \right)} \right] \times F_{eye}$$

3) The reviewer is correct at this point. This has been addressed in the revised manuscript. Thank you.

28. Page 8, line after Eq. (10): it is not clear what the author calls "eyepiece". In this line it seems the term designates the combination of lenses L3 and L4 ("The rays […] are guided through the eyepiece (lenses L3 and L4 in Fig. 1b and Fig. 2 respectively)"), but the used terminology, calling Fobj the focal length of L3 and Feye the focal length of L4 would lead think that what's called "eyepiece" is L4 (and L3 "objective", by the way). This should be clarified. In addition, the meanings of Fobj and Feye should be explained in the text and in the caption of fig. 2.

Correct. I would like to thank the reviewer for pointing out this misleading sentence. The designation of the used optical components is already described in Section 2.1. According to reviewer's remark No. 11, in the updated manuscript a clear statement is given, matching the objective lens as the optical layout (L3) (Page 5 line 13) and the eyepiece as the optical layout (L4) (Page 5 line 14), as those have been identified in Fig. 1b.

This has been further clarified at this point of the updated manuscript (Page 8 lines 7-8) by changing this sentence as follows:

"The rays collected by the IFF and the objective lens (L3 in Fig. 1b), are guided through the eyepiece lens (L4 in Fig. 1b), creating"

29. Page 8, second line after Eq. (10): "creating the second intermediate image plane at a distance Z3". It would be more precise to say "creating the final image of the entrance pupil at distance Z3"; there are not more images after this one.

Corrected according to reviewer's suggestion, stating that this is the final image of the entrance pupil created at distance $Z_3$ behind the eyepiece lens. Thank you.

30. Page 8: the paragraph starting after Eq. (13) "PMTs suffer from a non-uniform spatial response of their effective surface…" is crucial for the optical layout being described. This is the reason why one wants to create an image of the telescope aperture on the surface of the photodetector. This should be stressed and the idea be expressed earlier in the paper, perhaps appearing in the abstract and in the introduction.

The reviewer is gratefully acknowledged for pointing this issue. The idea of imaging the telescope aperture on the phototodetector has been emphasized earlier in the abstract and the introduction of the revised manuscript. More precisely, in the abstract section (page 1, line 3) this is now expressed as follows:

"For projecting the telescope aperture on the detection surface and avoiding its spatial inhomogeneities, an eyepiece lens has to be used in the lidar setup. The set of the derived equations, includes also the geometrical characteristics of that eyepiece lens."

While in the introduction section the following text has been inserted in the revised manuscript (page 3, line 32):

"The derived formulation includes also the characteristics of an eyepiece lens, which has to be used in a lidar setup in order to optically form the light collected by the telescope on the detector, homogeneously and with the same size, irrespectively of the height range of the scattered light."

31. Page 8, sentence starting on line 10: "At this place (distance $Z3$ behind the eyepiece), the image of the lidar beam does not move with the lidar distance, and the spatial intensity distribution over the PMTs active surface does not change".
Remarks:
1) What's the "lidar beam"? Maybe the author means the "laser beam". What is meant by "lidar distance"? "Distance to the lidar" would be more precise.
2) The sentence is anyway misleading: at $Z3$ behind the eyepiece (or lens L4) the optical system is not forming an image of the beam, but an image of the telescope aperture, therefore spreading the light coming from illuminated points in the atmosphere uniformly over the photodetector surface. Lidar systems that image the laser beam onto the photodetector surface suffer from the inhomogeneities mentioned earlier.

The reviewer is gratefully acknowledged for pointing out this sentence was misleading anyway. In the revised manuscript (page 9, line 20) this paragraph has been changed as follows:

"At this place (distance $Z_3$ behind the eyepiece lens (L4)), the system is forming the image of the telescope aperture, therefore spreading the light coming from illuminated points in the atmosphere uniformly over the photodetector surface (S2). In case that L4 was missing from the described optical setup, the photodetector should have been placed on the focal plane of the objective lens (L3), and the optical system would finally form there the image of the laser beam, inhomogeneously over the photodetector surface, introducing errors in the recorded lidar signals, as mentioned earlier."

32. Page 8, sentence starting on line 11: "In addition, an advantage of using makes the detection surface rather insensitive…" . Something is missing after "using".
The missing word "an eyepiece lens" has been added. Thank you.

33. Page 9, line 14: "are provided in the following paragraph (Section 3)". Remark: "are provided in Section 3" would suffice.

Corrected, as suggested by the reviewer.

34. Page 10, sentence starting on line 5: "and 180 mm above the optical axis". For the sake of generality it would be better "and 180 mm from the optical axis".

Corrected, as suggested by the reviewer.

35. Fig. 4: it is not clear what the 5 diagrams of fig. 4a correspond to. Do they correspond to rays coming from 10 different points, 5 in the far range and 5 in the near range, at different positions within the receiver field of view? Whatever they correspond to, it should be said both in the text and in the figure caption. Note as well that a and b are missing to identify figs. 4a and 4b.

Thank you. The reviewer is right. The explanation of this figure both in the caption and in the text was not sufficient. Following the reviewer's suggestion the following text has been inserted in the revised manuscript:

"The five spot diagrams demonstrated in Fig. 4a correspond to rays coming from five different positions (field points) within the receiver field of view. The field points have been calculated assuming an object height with radius equal to $\mathrm{LBD} \times Z + \frac{\mathrm{DL}}{2}$, where $Z$ is the atmospheric distance from the lidar."

Moreover, Fig. 4 has been updated, by inserting the characters (a) and (b), and the corresponding caption has been changed to:

"Figure 4: Spot diagrams of far (green; 10000 m) and near (blue; 257 m) range rays on the (a) front surface of the IFF from five different positions within the receiver field of view, and (b) PMT detector with 5 mm effective diameter (black circle)."

36. Page 10, sentence starting on line 22: "For example, ZEMAX simulations revealed that telescope's primary mirror is focusing the near and far field rays at different planes (Fig. 5)". This seems to be ascribed to aberration effects ("the inability of paraxial approximation to take into account all kinds of possible aberrations, in contrast to ZEMAX simulation"). However this simply results from the paraxial formula relating object and image positions (by the way, given in the first equation of equation block (17); in fact this effect seems to be taken into account in Section 7). A simple calculation shows that, for a thin lens of 600 mm focal length, the image of a point at 10000 m from the lens will be at 600.04 mm from the lens plane, while the image of a point at 257 m will be at 601.40 mm, the difference being 1.37 mm, very close to the 1.40 mm indicated in Fig. 5. So the effect seems to be explained by paraxial optics. As to the ensuing discussion on where the field stop should be placed, in my opinion it should be dropped: the displacement of the image point being a paraxial effect, it has already been implicitly taken into account in the previous developments.

I would like to thank the reviewer for correctly pointing this issue. According to his suggested comments the entire paragraph has been revised in the manuscript as follows:

"and (b) the inability of the paraxial approximation to take into account the sag of each lens surface to better model the refraction of off-axis rays, in contrast to ZEMAX simulation. However, the effect of lens defocus aberration seems to be taken into account by paraxial optics. For example, ZEMAX simulations

revealed that telescope's objective lens is focusing the near and far field rays at different planes (Fig. 5). The far field rays, are focused exactly on the focal plane of the telescope while the near field rays are focused in a plane with an axial shift of 1.4 mm on the optical axis. From the paraxial formula relating object and image positions (see Eq. 17 of Section 7) a simple calculation shows that, for a thin lens of 600 mm focal length, the image of a point at 10000 m from the lens will be at 600.04 mm from the lens plane, while the image of a point at 257 m will be at 601.40 mm, the difference being 1.36 mm, very close to the 1.40 mm revealed by ZEMAX simulations (Fig. 5)."

37. I don't understand Section 4. It starts by saying that "An axial shift of $Z_1$ is leading to a shift of the $A_{IFF}$", then that the shift is produced by displacing L3. But in my understanding (although not completely clear due to some possible notation inconsistences, see remark No. 21) $A_{IFF}$ is the acceptance angle of the interference filter, so I don't see how it can be affected by the shift of a lens. Even if $A_{IFF}$ refers to the angle with respect to the axis of the rays exiting L2, it cannot be changed by the displacement of an element coming after it.

The reviewer is gratefully acknowledged for this comment. I agree with him that the first two sentences of Section 4 are misleading. The aforementioned have been changed as follows:

"An axial shift of IFF, will introduce a change of the distance $Z_1$ which is leading to a shift of the $A_{IFF}$ (Eq. 8), affecting the RFOV of the system (Eq. 2), and resulting consequently to a change of the DFO (Eq. 1). Thus, in order to identify the contribution of a possible axial shift of the IFF (presented in Fig. 1b) to the DFO, an iterative based Monte Carlo method, has been applied."

38. I'm not sure section 5 is relevant, as, in my opinion, its conclusions are implicit in the considerations of the previous sections: the field stop, together with the telescope focal length, determines the receiver field of view, and all the rays reaching the telescope aperture coming from illuminated points in the full field of view zone, as defined in fig. 2 of Stelmaszczyk et al. 2005, will pass through the field stop. Moreover there are some inaccuracies:

a) Page 11, line 20: "assuming that the primary mirror of the telescope is an ideal thin positive lens". Again this seems to assume a Newtonian telescope, which is not always the case.

I agree with the reviewer. Thank you. In the revised manuscript this has been already corrected as follows:

"According to the Gaussian lens formula and treating a telescope as an ideal thin positive lens, one can get"

b) Page 12, line 7: "The image height $y_i$ becomes zero for an object projected from infinity." This is true for objects at finite distance from the axis, but, without further clarifications, the sentence can contradict Eqs. (18) and (19), where, even if, $z_b \rightarrow \infty$ $x_i$ and $y_i$ do not tend to 0. This is because for divergent and/or tilted beams, farther points in the beam are at farther distances of the axis (i.e. $\frac{x_b}{z_b}$ and $\frac{y_b}{z_b}$ remain constant).

The reviewer is gratefully acknowledged for pointing this inaccuracy. In the revised manuscript this has been clarified as follows:

"The image height $y_i$ becomes zero for an object which is projected from infinity ($z_b \rightarrow \infty$ ) and has finite distance from the optical axis. However, this is not always the case, since for divergent and/or tilted beams, where farther points in the beam are at farther distances of the axis (i.e. $\frac{x_b}{z_b}$ and $\frac{y_b}{z_b}$ remain constant), the image width and its height from the optical axis are not zero. The far range points of the object are projected almost in the centre of the field stop on optical axis, leading the smallest area of illumination on the FOVD."

39. Page 12, sentence starting on line 13: "from the emitted laser beam to its projection on the photomultiplier": this may lead to think that the laser beam is imaged onto the photomultiplier surface, which is not the case in the considered setup (see remark No. 29)

Once again I would like to thank the reviewer for pointing this inaccuracy. In the revised manuscript this has been corrected as follows:

"from the emitted laser beam to imaging the entrance pupil of the telescope onto the surface of the photodetector."

40. Page 12, sentence starting on line 16: "The usage of IFF with small bandwidth for background suppression is limited by their small acceptance angle in near field". The acceptance angle of an interference filter is independent of where the rays originate, whether in the near range or in the far range. I think that "in the near field" should be removed.

Correct. The sentence "in the near field" has been removed in the revised manuscript.